# Iron mineral dissolution releases iron and associated organic carbon during permafrost thaw

Monique S. Patzner [1], Carsten W. Mueller [2,3], Miroslava Malusova[1], Moritz Baur[1], Verena Nikeleit[1], Thomas Scholten [4], Carmen Hoeschen [2], James M. Byrne[1,5], Thomas Borch [6], Andreas Kappler [1] & Casey Bryce[1,5 ✉]

It has been shown that reactive soil minerals, specifically iron(III) (oxyhydr)oxides, can trap organic carbon in soils overlying intact permafrost, and may limit carbon mobilization and degradation as it is observed in other environments. However, the use of iron(III)-bearing minerals as terminal electron acceptors in permafrost environments, and thus their stability and capacity to prevent carbon mobilization during permafrost thaw, is poorly understood. We have followed the dynamic interactions between iron and carbon using a space-for-time approach across a thaw gradient in Abisko (Sweden), where wetlands are expanding rapidly due to permafrost thaw. We show through bulk (selective extractions, EXAFS) and nanoscale analysis (correlative SEM and nanoSIMS) that organic carbon is bound to reactive Fe primarily in the transition between organic and mineral horizons in palsa underlain by intact permafrost (41.8 ± 10.8 mg carbon per g soil, 9.9 to 14.8% of total soil organic carbon). During permafrost thaw, water-logging and $O_2$ limitation lead to reducing conditions and an increase in abundance of Fe(III)-reducing bacteria which favor mineral dissolution and drive mobilization of both iron and carbon along the thaw gradient. By providing a terminal electron acceptor, this rusty carbon sink is effectively destroyed along the thaw gradient and cannot prevent carbon release with thaw.

[1] Geomicrobiology, Center for Applied Geosciences, University of Tuebingen, Tuebingen, Germany. [2] Chair of Soil Science, Technical University Muenchen, Freising, Germany. [3] Department of Geosciences and Natural Resource Management, University of Copenhagen, Copenhagen, Denmark. [4] Chair of Soil Science and Geomorphology, University of Tuebingen, Tuebingen, Germany. [5] School of Earth Sciences, University of Bristol, Bristol, UK. [6] Department of Soil & Crop Sciences and Department of Chemistry, Colorado State University, Fort Collins, CO, USA. ✉email: casey.bryce@bristol.ac.uk

The release of vast amounts of organic carbon during thawing of high-latitude permafrost is an emerging issue of global concern. Yet, the extent of greenhouse gas emissions from permafrost thawing remains unpredictable due to knowledge gaps related to controls on the fate of carbon in permafrost soils[1]. The mobility, lability, and bioavailability of organic carbon is determined by a number of inter-connected physico-biogeochemical parameters and processes. One such parameter is the presence of reactive iron (Fe) minerals (defined here as iron minerals that are reductively dissolved by the chemical reductant sodium dithionite, e.g. ferrihydrite or goethite), which are known to stabilize organic carbon by sorption and co-precipitation[2–5]. Fe-bound carbon can be protected by soil structural conditions (such as aggregate formation, macro-scale shifts in fluid flow paths), thus be less accessible to decomposer organisms[6,7]. At the same time, oxygen ($O_2$) diffusion is hindered further favoring soil organic matter (OM) preservation rather than decomposition[6,7]. Thus, Fe–OM associations are thought to significantly influence long-term carbon storage in numerous environments[5,8–10].

Several studies already identified poorly crystalline Fe-OM associations in the field or produced them in the lab, and demonstrated that they are resistant to microbial or chemical reduction[11–13]. The inventory of reactive iron minerals in humid climates is highly dynamic as they precipitate and dissolve in response to changing redox conditions. Mineral soil slurry incubations previously showed OM protection by iron only under static oxic conditions[14]. However, iron(III) mineral reduction and dissolution under oxygen limitation led to anaerobic mineralization of dissolved OM and soil OM by 74% and 32–41%, respectively[14]. When mineral dissolution occurs, iron and carbon mobilization, increased carbon lability/bioavailability, and increased gaseous carbon loss as $CO_2$ and $CH_4$ follow (catalyzed by heterotrophic and methanogenic microorganisms)[15–19]. The extent to which the formation and dissolution of reactive Fe phases contribute to soil OM persistence in redox-dynamic permafrost soils remains unknown. Despite the importance of iron minerals for carbon storage, we have little understanding of the presence of this so called rusty carbon sink in organic-rich permafrost peatlands and even less knowledge of how it will respond to changing redox conditions associated with permafrost thaw.

In order to address this knowledge gap, we examined peatland soils collected along a thaw gradient at Stordalen mire (Abisko, Sweden). For this study, three thaw stages were defined, based on possible shifts in vegetation and hydrology, as has been done previously[20–23], and known changes in microbial ecology[24,25] (Fig. 1 and Supplementary Fig. 1). These are: (1) desiccating palsa underlain by permafrost, (2) intermediately thawed bog, and (3) fully thawed fen. Desiccating palsa sites are mainly dry and oxic[21]. As the permafrost thaws, the raised, dry ombrotrophic palsas collapse, causing enlargement of semiwet and wet ombrotrophic bog areas with continuously frozen soil underneath[21,22]. With continued thawing, the bog areas decrease and minerotrophic fen areas expand with complete water saturation and thus even more reducing conditions than in the bog[20–22,26]. To be able to use a space-for-time approach, the habitats are ordered following a temporal succession of apparent time since onset of thaw, as has been done by Hodgkins et al.[21].

We collected cores from locations deemed to be most representative of the three thaw stages and analyzed iron–carbon associations in three different layers within the cores defined by geochemical stratification, as has been done previously[27]. Cores were split into (1) organic horizon, (2) transition zone, interface between the two soil horizons, which can be locally very sharply defined (3–5 cm), but also reach over 20 cm thick[27], and (3) mineral horizon (Supplementary Fig. 1). We analyzed the solid phase by selective extractions to determine different iron phases such as reactive iron (dithionite citrate extractable), poorly crystalline iron (hydroxylamine-hydrochloric acid (HCl)), colloidal and OM-chelated iron (sodium pyrophosphate), and more crystalline iron phases such as poorly reactive sheet silicate Fe or Fe sulphur (S) species (6 M HCl). The selective extractions were simultaneously used to determine iron-bound carbon by measurement of dissolved organic carbon (DOC) in the sodium dithionite citrate and sodium pyrophosphate extractions. Extended X-ray absorption fine structure (EXAFS) analysis was performed to support extraction data and further determine the presence of phases such as Fe-containing clays which were not extracted by the selective dissolutions. Correlative scanning electron microscopy (SEM) and nanoscale secondary ion mass spectrometry (nanoSIMS) were used to visualize iron–carbon (C) associations in the fine fraction of the different soil layers. This was complemented by geochemical analysis of the porewater to determine potential electron acceptors and donors, and to track iron and carbon release from the solid into the liquid phase. Additionally, Fe(III)-reducing bacteria which are the driving force behind iron mineral reduction under anoxic conditions were quantified by a growth-dependent approach. We found that reactive Fe can bind organic carbon primarily in palsa soils underlain by intact permafrost. Water-logging and $O_2$ limitation during permafrost thaw lead to reducing conditions and an increasing abundance of Fe(III)-reducing bacteria. By using the reactive Fe as terminal electron acceptor, they favor mineral dissolution and drive mobilization of both iron and carbon along the thaw gradient. Thus, the rusty carbon sink is destroyed and cannot preserve organic carbon from carbon mineralization with permafrost thaw.

## Results and discussion

**Thaw increases aqueous $Fe^{2+}$ and DOC.** Along the thaw gradient, the aqueous $Fe^{2+}$ in the porewater increased from average concentrations of $0.02 \pm 0.01$ mM in the palsa to up to $1.6 \pm 0.3$ mM in the fen transition zone (Fig. 2). This correlates with an increase in average DOC from $19.7 \pm 0.8$ mg/L in the palsa to $102.1 \pm 14.1$ mg/L in the fen area. In the bog porewater, acetate ($0.6 \pm 0.1$ mM) was measurable in the transition zone and mineral horizon, whereas lactate ($0.8 \pm 0.02$ mM) was only found in the transition zone. (Fig. 2). At the fen site, lactate and acetate were detected throughout the depth profile ($0.2 \pm 0.1$ mM). An additional peak in acetate ($2.3 \pm 0.01$ mM), propionate ($0.8 \pm 0.02$ mM), and butyrate ($0.2 \pm 0.01$ mM) was observed in the transition zone of the fen (Fig. 2). The appearance of butyrate and propionate in the fen porewater could be an indicator for ongoing microbial processes such as fermentation and methanogenesis in the more water-logged and thus more reduced fen soils. This is in line with observations from previous studies that highest methane emissions occur in the fen[24]. Our porewater data along the thaw gradient clearly show an increase in aqueous $Fe^{2+}$ and more labile organic carbon. It also highlights unique biogeochemical processes in the transition zone leading to the consumption or accumulation of fatty acids in this layer. The presence of more labile organic carbon in the porewater is consistent with previous work[21]. The palsa to bog catchment was described previously to have no other extrinsic sources of Fe or DOC[28]. The fen catchment, however, is hydrologically influenced by palsa to bog flow and by surrounding surface water bodies (ponds, river, and lake)[28]. Analysis of this surface water showed average DOC concentrations of $24.87 \pm 6.68$ mg/L and average aqueous $Fe^{2+}$ concentrations of $0.02 \pm 0.02$ mg/L. Thus, these extrinsic surface water sources cannot explain the high Fe and DOC measured in the fen.

**The abundance of Fe(III)-reducing bacteria increases with thaw.** The trend in increasing aqueous $Fe^{2+}$ and DOC concentrations observed across the thaw gradient goes hand in hand with a

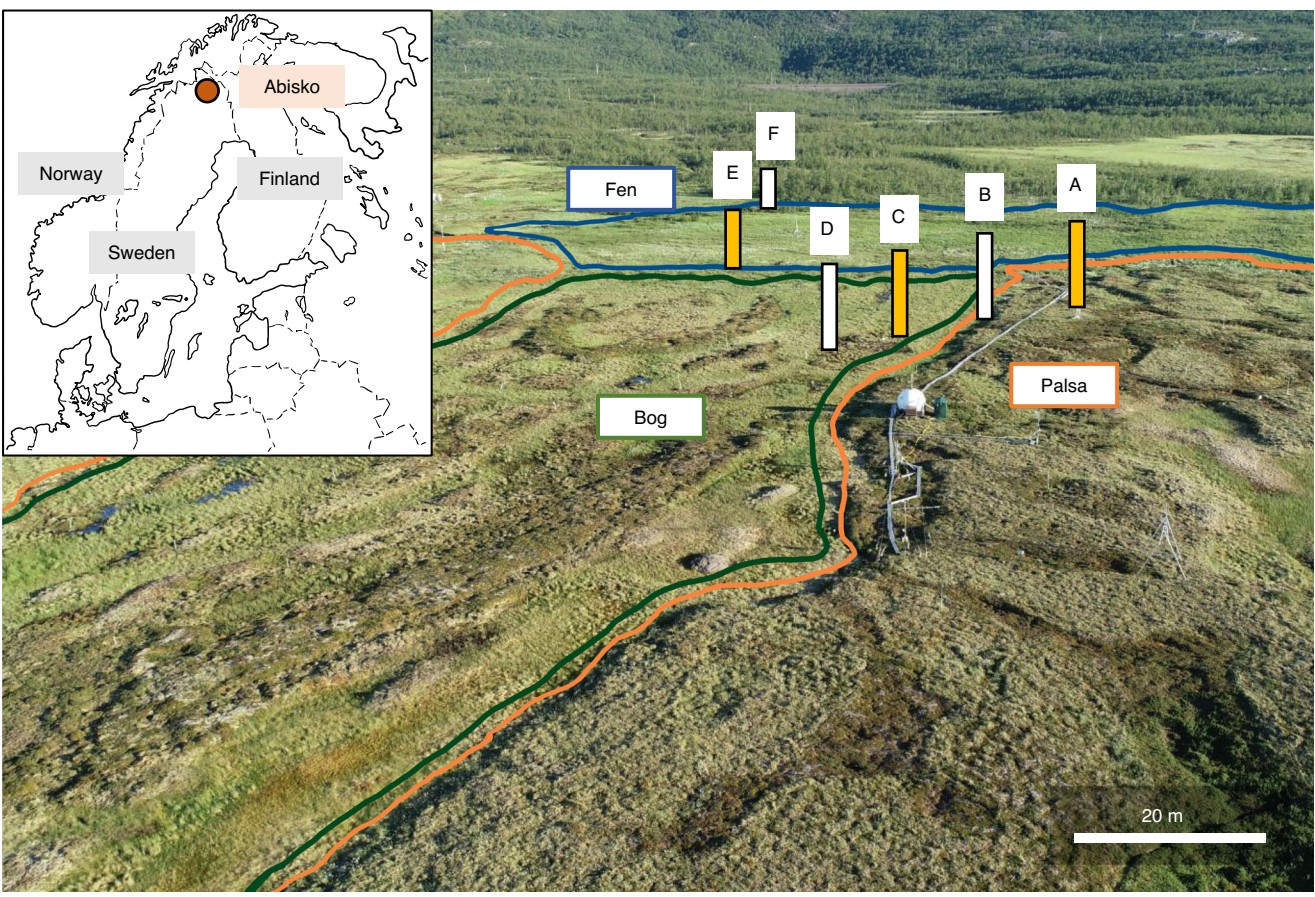

**Fig. 1 Field site Stordalen mire close to Abisko in the North of Sweden.** The three main thaw stages are (1) palsa (marked in orange), (2) bog (in green), and (3) fen (in blue). The positions of the three cores analyzed in detail within 3–4 days of collection in 2018, which represent all three thaw stages, are shown in yellow. Additional cores (shown in white) were taken in 2018 and analyzed after 7 months of incubation at 4 °C (Supplementary Figs. 1 and 3). Data for further replicates, taken in 2017 and 2019, are provided in the SI (Supplementary Figs. 1–5).

significant increase in abundance of Fe(III)-reducing bacteria in the organic horizon from palsa to fen (unpaired $t$-test, $N = 7$, $\alpha = 0.05$, $p = 0.0001$) and a significant increase in the transition zone from palsa to fen (unpaired $t$-test, $N = 7$, $\alpha = 0.05$, $p = 0.0001$). In the mineral horizon at all three thaw stages, no difference in abundance of Fe(III)-reducing cells was observed ($\sim 2 \times 10^3$ cells per g soil). In the palsa organic horizon, $2.4 \times 10^2$ cells per g soil were found, in comparison to $2.6 \times 10^4$ cells per g soil in the organic horizon in the fen. In the transition zone of palsa, $2.4 \times 10^3$ cells per g soil were determined. Whereas, in the fen transition zone, the highest abundance of Fe(III)-reducing bacteria with $3.1 \times 10^5$ cells per g soil was observed. The abundance of Fe(III)-reducing bacteria was determined via growth and Fe(III) reduction by most probable numbers (MPNs). Further isolation efforts from the fen yielded a microorganism with 99% similarity on the 16S rRNA level to *Cupriavidus metallidurans* (Proteobacteria). The isolate was able to reduce ferrihydrite to Fe(II) whilst simultaneously consuming lactate and producing acetate (Supplementary Fig. 6). It did not utilize acetate. Fe-associated organic carbon could further enhance or lower reduction rates of ferrihydrite (depending on the OC:Fe ratios), but is unlikely to prevent mineral dissolution[29]. *C. metallidurans* strains are known to be extremely metabolically flexible and can utilize lactate fermentation under anoxic conditions[30]. They are also highly tolerant of toxic metals[30]. Although Fe(III) reduction is usually thought of as a dissimilatory process, in some peatlands, Fe(III) reduction coupled to fermentative metabolisms have been observed to be more common[31]. This could be one explanation for the acetate peak in the fen transition zone and increasing Fe(II) concentrations in the fen.

Woodcroft et al.[25] describes evidence of lactate fermentation along this thaw gradient and found Proteobacteria as one of primary lactate metabolizers in the fen. Our growth-dependent quantification is suggestive of increased microbially driven Fe(III) mineral dissolution along the thaw gradient (Fig. 2).

**Reactive iron minerals are dissolved along the thaw gradient.** In order to determine whether the observed increases in DOC were related to increased Fe(III) mineral reduction and dissolution, we quantified the amount of organic carbon associated with the reactive iron minerals along the thaw gradient by applying the citrate–dithionite iron reduction method[5,32,33]. This method simultaneously dissolves all reactive solid iron phases and releases the organic carbon associated with these minerals into solution. The extraction is performed at circumneutral pH to prevent hydrolysis of OM as well as its protonation and re-adsorption onto the remaining solid phases and thus its precipitation. A control experiment was conducted at the same pH with equivalent ionic strength (sodium chloride instead of the reducing agent sodium dithionite). The organic carbon which is released in this control is not associated with the reactive iron minerals and was therefore subtracted from the amount of carbon released from the dithionite-citrate extractions as previously described[5] (Supplementary Table 1). Well-known issues with these extractions and a discussion of how we have overcome these are included in the Supplementary Information (Supplementary Method 1, Supplementary Figs. 7 and 8). Additionally, we performed a

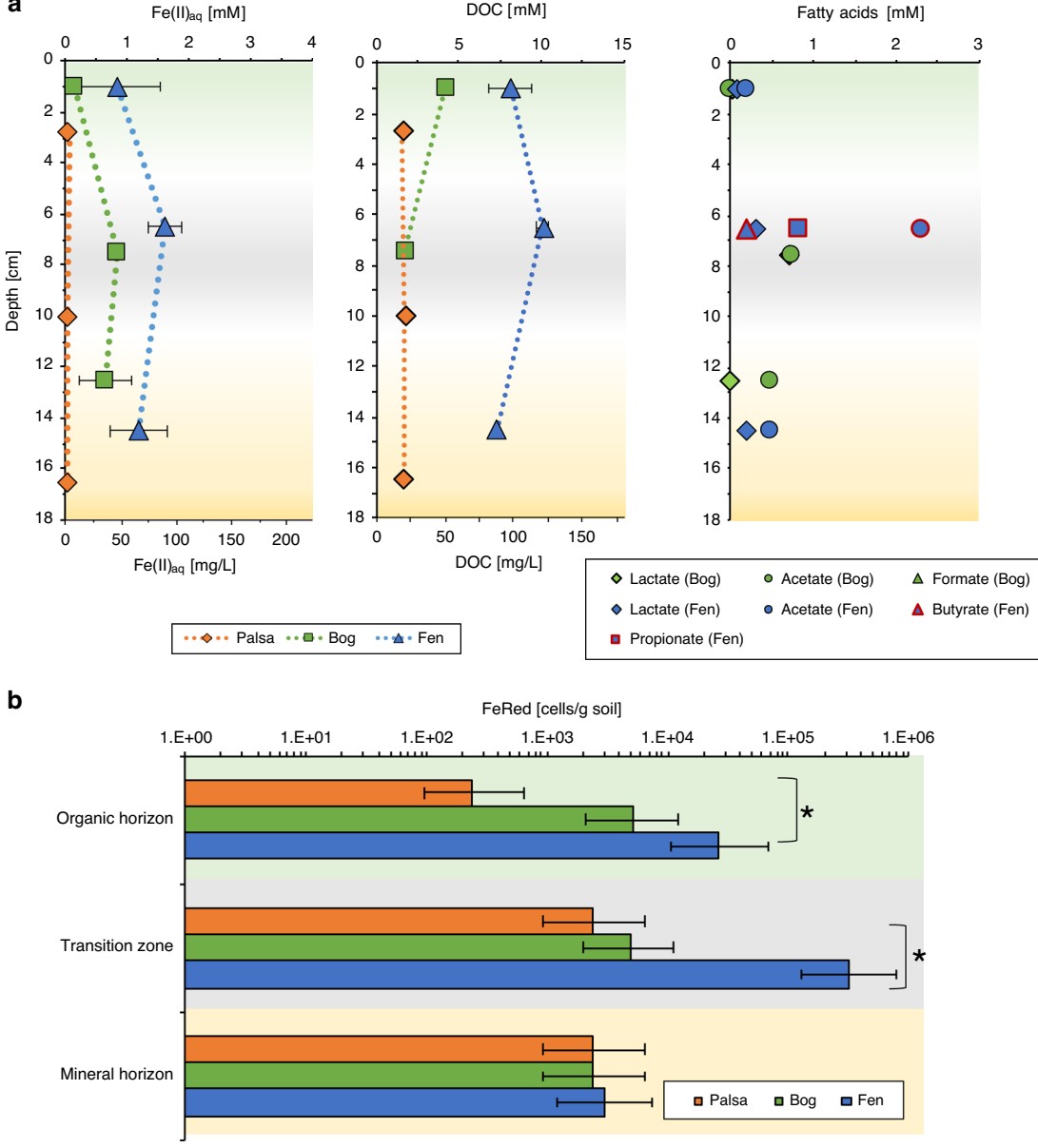

**Fig. 2 Increasing iron and dissolved organic carbon concentrations accompanied by an increase in abundance of Fe(III)-reducing bacteria along the permafrost thaw gardient. a** Porewater geochemical analysis of the cores Palsa A, Bog C, and Fen E and **b** most probable number quantifications of Fe(III)-reducing bacteria (FeRed) in the solid phase of the cores Palsa A, Bog C, and Fen E along the thaw gradient. Red marks the high acetate concentrations in the fen, in comparison to the bog, and the additional fatty acids butyrate and propionate, only detected in the fen. The error bars of the porewater data represent triplicate measurements. The error bars of the most probable number estimations of Fe(III)-reducers represent seven replicate analyses and indicate lower and upper limits of the 95% confidence intervals; * marks significant difference between FeRed in palsa and fen, organic horizon, and transition zone (unpaired $t$-test, $N = 7$, $\alpha = 0.05$, $p = 0.0001$). The green background marks the organic horizon, gray the transition zone, and yellow the mineral horizon. The backgrounds for the porewater geochemistry are shaded due to the fact that this represents three cores, one core per thaw stage, with different horizon depths.

sodium hydroxylamine-HCl extraction (pH <2) to target the poorly crystalline iron minerals, a sodium pyrophosphate extraction (pH 10) to extract colloidal or OM-chelated iron, and a 6 M HCl extraction to obtain more crystalline iron phases of the soil layers (referred to as mg Fe(tot) per g soil) (Fig. 3). It should be noted that the total amount of iron per dry weight in the layers is different along the thaw gradient due to different redox-driven biogeochemical cycles in the three different thaw stages resulting in loss of total soil organic carbon and thus in an increasing abundance of the mineral material present in the active layer (Supplementary Figs. 1–4 for spatial and horizontal variance in

replicate cores). Stocks (mg/cm$^2$) were calculated, but due to slight compaction, mainly for the organic horizons, not reported in the main text (Supplementary Fig. 9). In the following, only the data from cores Palsa A, Bog C, and Fen E are discussed (Figs. 1, 3, and 4), but observed trends are supported by further analyses conducted on cores analyzed in the same manner, but collected in 2019 (Palsa a, Bog c, and Fen e) (Supplementary Fig. 2), as well as cores collected at the same time as Palsa A, Bog C, and Fen E in 2018 but stored for a longer period (Palsa B, Bog D, Fen F) (Supplementary Fig. 3). We also observed similar trends in triplicate cores from each thaw stage, collected in 2017 and analyzed

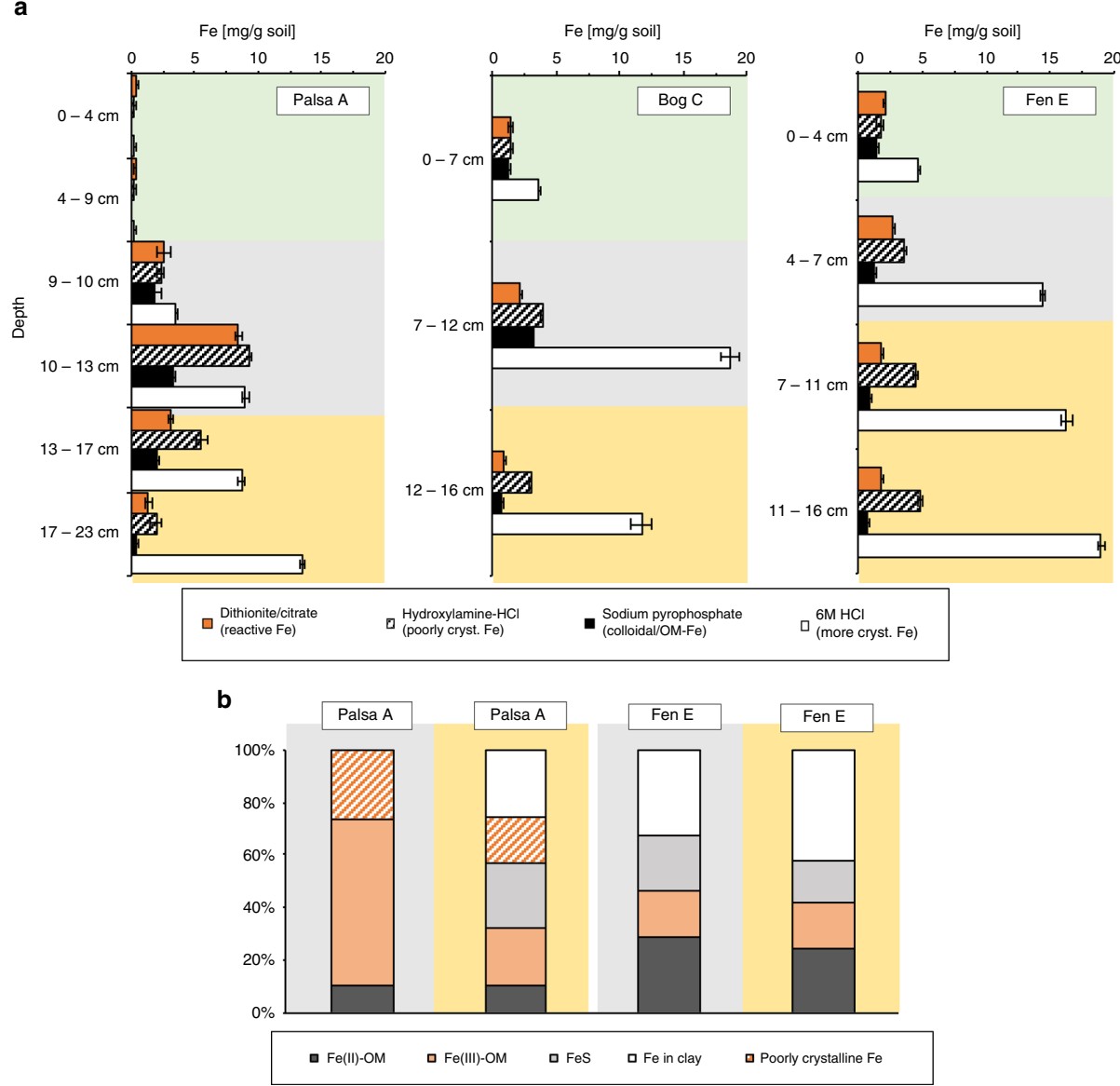

**Fig. 3 Reactive Fe along the permafrost thaw gradient.** Fe speciation was determined along the thaw gradient by selective extractions (**a**) and extended X-ray absorption fine structure (EXAFS) (**b**). The reactive iron mineral fraction (dithionite citrate extractable) [mg iron (Fe) per g soil] was quantified in the different layers, control corrected by a sodium chloride bicarbonate extraction with the same ionic strength and pH (Supplementary Table 1), and compared to the poorly crystalline Fe (hydroxylamine-hydrochloric acid (HCl) extractable Fe), the more crystalline Fe (6 M HCl extractable Fe, referred to mg Fe(tot) per g soil in the text) and to the colloidal and/or organic matter (OM)-chelated Fe (sodium pyrophosphate extractable Fe). Please note the differences in the scale of the y-axis due to variable thickness of each soil layer along the thaw gradient. The green background marks the organic horizon, gray the transition zone, and yellow the mineral horizon. Error bars of all extractions represent duplicate extractions of each layer per thaw stage, except for the dithionite citrate extractable iron which represents a combined standard deviation of sodium chloride bicarbonate extractable iron and dithionite/citrate extractable iron (not control corrected). EXAFS results of the transition zone and the mineral horizon of the two-end members palsa and fen show loss of the poorly crystalline Fe (reference probe: 2-line ferrihydrite), the decrease in OM-chelated Fe (reference probes: Fe(II)-citrate and Fe(III)-citrate), the increase of Fe in clays (reference probes: natural nontronite and ferrosmectite), and Fe sulfur species (reference probe: mackinawite) with depth and along the thaw gradient. Absolute values are reported in Supplementary Table 1.

with different but comparable methods (Supplementary Figs. 4 and 5). The collected cores capture the three thaw stages over 3 years (2017, 2018, and 2019) (Supplementary Fig. 1) and show that the trends of Fe and organic carbon along the thaw gradient are robust.

In the transition zone of the palsa, the reducible iron mineral fraction was 72.9–93.9% of the total extractable iron (2.6 ± 0.6 to 8.44 ± 0.2 mg reactive Fe per g soil in comparison to 3.5 ± 0.1 to 9.0 ± 0.3 mg Fe(tot) per g soil) (Fig. 3 and Supplementary Table 1). We suggest that this is driven by Fe(III) reduction in

deeper layers leading to mobilization of $Fe^{2+}_{aq}$. This can be subsequently re-oxidized close to the interface between the organic horizon and mineral horizon by $O_2$ which diffuses from the surface. Oxidation will result in precipitation as Fe(III) oxyhydroxide minerals at this transition, as has been previously suggested for boreal peat soils[34]. The amount of reactive iron minerals in the transition zone then decreased to 11.1% of the total extractable iron in the bog (2.1 ± 0.1 mg reactive Fe per g soil in comparison to 18.7 ± 0.7 mg Fe(tot) per g soil) and to 18.3% of the extractable iron in the fen (2.6 ± 0.03 mg reactive Fe per g soil

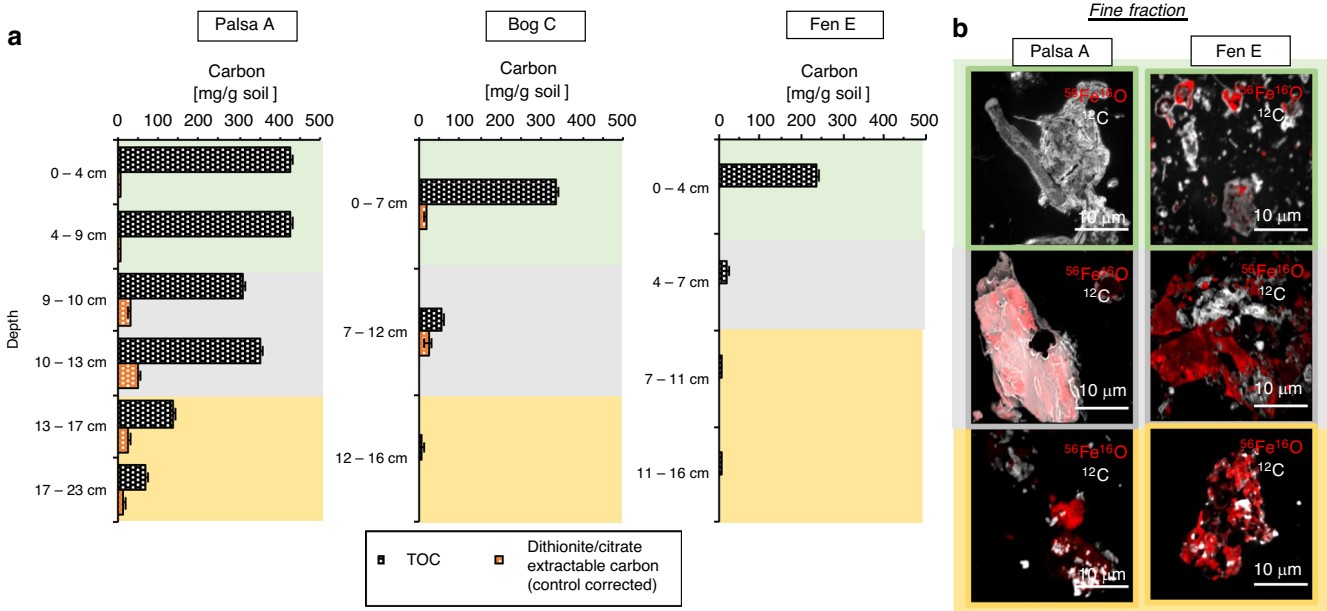

**Fig. 4 Reactive Fe-bound organic carbon along the permafrost thaw gradient.** Iron (Fe) and carbon (C) associations were determined along the thaw gradient by bulk (**b**) and fine fraction analysis (**a**). **a** Carbon bound by reactive iron minerals along the thaw gradient. The carbon which dislodged from the soil during the reductive dissolution of reactive iron oxides (orange) is shown in comparison to the total organic carbon determined via combustion (black grids, labeled as total organic carbon (TOC)). Dithionite–citrate extractable carbon is control-corrected by subtracting the measured dissolved organic carbon (DOC) content of a citrate solution and the measured DOC value from the sodium chloride (NaCl) control experiment. The NaCl control (same ionic strength and same pH as the sodium dithionite citrate extraction) shows negligible carbon release (Supplementary Table 1). Errors of the TOC indicate the range of duplicate analyses of each layer in each thaw stage. Errors of the dithionite/citrate extractable carbon (control corrected) represent a combined standard deviation of sodium chloride bicarbonate extractable OC, citrate blank, and dithionite/citrate extractable OC (not control corrected). **b** High spatial resolution analysis of iron–carbon associations by nanoSIMS along the thaw gradient (two end-members palsa (left) and fen (right)). The strong spatial association of carbon to iron (III) minerals could only be observed in the palsa transition zone. The other fine fractions showed organic-free iron minerals. For the two end-members palsa and fen, four particles of the fine fractions of each layer were analyzed by nanoSIMS, all showing the same spatial distribution of Fe and C as shown by these six representatives (see also Supplementary Fig. 11). The green background marks the organic horizon (**b**, upper images), gray the transition zone (**b**, middle images), and yellow the mineral horizon (**b**, lower images).

in comparison to $14.5 \pm 0.2$ mg Fe(tot) per g soil) (Fig. 3 and Supplementary Table 1). In the mineral horizons along the thaw gradient, a loss of the reactive iron minerals was also observed, likely due to more reduced conditions favoring Fe(III) mineral reduction at deeper depths. Reactive iron in the palsa mineral layer was 10.0–36.6% of the total extractable iron ($3.2 \pm 0.2$ to $1.4 \pm 0.2$ mg reactive Fe per g soil in comparison to $8.7 \pm 0.3$ to $13.5 \pm 0.2$ mg Fe(tot) per g soil) and declined to 7.5% of the total extractable iron in the bog ($0.9 \pm 0.1$ mg reactive Fe per g soil in comparison to $11.7 \pm 0.8$ mg Fe(tot) per g soil) and 9.0–10.7% of the total iron in the fen ($1.8 \pm 0.04$ to $1.7 \pm 0.04$ mg reactive Fe per g soil in comparison to $16.3 \pm 0.4$ to $19.01 \pm 0.25$ mg Fe(tot) per g soil) (Fig. 3 and Supplementary Table 1). This loss of reactive iron in the transition zone and mineral horizon was also confirmed by the hydroxylamine-HCl extraction and iron speciation by EXAFS (Fig. 3). The abundance of colloidal and/or OM-complexed iron (defined as sodium pyrophosphate extractable iron) also decreased along the thaw gradient, giving way to an increasing relative atom percent-based abundance of Fe-bearing clays (Fig. 3). This observation is consistent with increasing aluminum concentrations in the extracts (Supplementary Fig. 10). The iron content in the organic layer increased from almost no iron in the palsa ($0.2 \pm 0.02$ mg Fe(tot) per g soil, all poorly crystalline) to $4.7 \pm 0.01$ mg Fe(tot) per g soil (with 43.4% of the total extractable iron being reactive iron) in the organic layer of the fen (Fig. 3). This might again be driven by Fe(III) reduction in deeper layers leading to mobilization of $Fe^{2+}_{aq}$, which might precipitate now close to the water table.

**Dissolution of reactive iron minerals releases associated organic carbon.** With the dissolution of reactive iron minerals along the thaw gradient, the mineral-associated carbon was mobilized. Organic carbon bound to the reactive iron phases is primarily found in the transition zone and the mineral horizon of the palsa, as well as in the transition zone of the bog (Fig. 4 and Supplementary Table 1). About 9.9–14.8% of the total organic carbon (TOC) ($31.0 \pm 0.7$ to $52.5 \pm 0.1$ mg organic carbon bound to reactive Fe per g soil) was released by reductive dissolution of reactive iron minerals in the palsa transition (Fig. 4 and Supplementary Table 1). In the palsa mineral horizon, 18.7–20.1% of the TOC ($13.6 \pm 0.42$ mg organic carbon bound to reactive Fe per g soil in comparison to $72.7 \pm 0.29$ to $136.1 \pm 0.2$ mg total soil organic carbon per g soil) was released. In the transition zone of the bog, 39.4% of the TOC ($22.7 \pm 8.6$ mg organic carbon bound to reactive Fe per g soil in comparison to $57.5 \pm 0.4$ mg total soil organic carbon per g soil) was associated with iron minerals (Fig. 4 and Supplementary Table 1). However, it should be noted that the total amount of carbon was less in these samples ($57.5 \pm 0.4$ mg total soil organic carbon per g soil) when compared to the palsa transition zone ($312.1 \pm 0.3$ to $354.7 \pm 0.04$ mg total soil organic carbon per g soil) due to total carbon loss along the thaw gradient. Highest total amounts of carbon bound by the reactive iron were therefore found in the palsa transition zone with an average of $41.8 \pm 10.8$ mg per g soil. This is also supported by the strong spatial association of C with Fe minerals in the fine fraction of this transition zone observed by nanoSIMS analysis (Fig. 4 and Supplementary Fig. 11), as has been also previously shown

for intact permafrost soils[35]. The maximum mass ratio of organic carbon to iron of 0.22, based on the maximal sorption capacity of reactive iron oxides for natural OM[4,36], as has been previously done to further characterize Fe–C associations in such systems[37,38], was exceeded for reactive Fe-associated organic carbon:reactive Fe, determined via the sodium dithionite citrate extraction, in the bulk palsa soil layers (organic/transition/mineral), in the bulk bog organic horizon, and in the bulk bog transition zone ($8.59 \pm 3.0$ OC:Fe (wt:wt)) (Supplementary Table 1). This suggests co-precipitation and/or chelation of organic compounds also in the bulk sample which can generate structures with OC:Fe ratios above 0.22, as shown in other studies[4]. Existence of such structures is consistent with our high sodium pyrophosphate extractable iron values.

**Implications for stability of Fe-associated carbon.** Carbon binding to reactive iron minerals in the palsa area is consistent with previous observations in permafrost regions of the Qinghai-Tibet Plateau[33] where Fe-associated organic carbon represents, on average, $19.5 \pm 12.3\%$ of the total soil organic carbon pool in the upper 30 cm of permafrost soils throughout the year. Assuming a carbon pool of $191.29 \times 10^{15}$ g carbon in the active layer (0–30 cm depth)[39] in northern permafrost regions, we suggest that $13.39 \times 10^{15}$ to $38.26 \times 10^{15}$ g carbon could potentially be bound to reactive iron in permafrost soils. The lower estimate assumes, based on our data, an average of 7% of TOC is bound to reactive iron in active layers underlain by intact permafrost (Supplementary Table 1, average mg C bound to reactive Fe per g soil in comparison to average mg TOC per g soil in %). The higher estimate assumes a maximum of 20% of TOC is bound to reactive iron, based on our data (Supplementary Table 1) and Mu et al.[33]. This Fe-bound carbon stock is equivalent to approximately 2–5% of the amount of carbon which is currently present in the atmosphere which is equivalent to between 2 and 5 times the amount of carbon released yearly through anthropogenic fossil fuel emissions. Fisher et al. even showed evidence for only partial dissolution of the Fe phases and associated organic carbon by the dithionite–citrate approach, indicating that carbon sequestration by Fe minerals is likely to be more significant than currently estimated[40]. However, this estimate does not account for carbon sequestration by biomass in the fen or deeper layers. Stock calculations show a higher Fe-associated OC fraction in mineral horizon than in the palsa transition zone, which could indicate that OC sequestration by reactive Fe minerals is likely to be more significant in deeper layers than currently estimated (Supplementary Fig. 9). It is therefore crucial to further determine the amount of iron in deeper layers, the amount of carbon bound to reactive iron minerals in numerous permafrost environments, and the lability/bioavailability of this carbon following its release.

Our space-for-time approach reveals, for the first time, how we may expect the dynamics of this rusty carbon sink to respond to progressive climate change. This study suggests that, as soon as the conditions in permafrost peatlands become water-logged, the reactive iron minerals are reduced, probably by Fe(III)-reducing bacteria, and dissolved iron and associated organic carbon are released into the surrounding porewater. As has been stated previously, reactive Fe phases require their own physiochemical protection in order to contribute to OM persistence[14] and the presence of OM itself can both increase or decrease the stability of Fe(III) minerals depending on the geochemical conditions. High C/Fe ratios, shown to enhance the extent of bioreduction compared to pure reactive iron mineral phases[29], could be one explanation as to why the reactive Fe phases present in these soils were not resistant to mineral dissolution with permafrost thaw. High C/Fe ratios were thought to enhance reduction rates[29] due

to the function of carbon as electron shuttle[41], its role as a strong ligand for Fe complexation[42], or its importance for particle aggregation[43]. Additionally, co-association of aluminum and iron with OM could have further made this rusty carbon sink more susceptible to reductive dissolution[44]. On the other hand, reduction-resistant surface coatings, embedding in a composite aggregate structure[6,13], or higher Fe mineral crystallinity[45] could prevent mineral dissolution.

Chen et al.[14] showed the formation of more crystalline Fe phases under $O_2$ fluctuations and suggested a substantial loss of co-precipitated OM under repeated redox fluctuations with a remaining Fe-bound fraction of OM resistant to reductive dissolution. In the bog thaw stage, redox fluctuations may have led to a core Fe-MAOM structure[14] (Fe mineral-associated OM) but constant reducing conditions in the fen caused a complete loss of Fe-bound carbon in the solid phase.

It should be kept in mind that it is not only the released carbon that can directly contribute to greenhouse gas emissions. The reduction of Fe(III) itself will also contribute to $CO_2$ emissions since it is directly coupled to the oxidation and mineralization of organic carbon. On the other hand, since Fe(III) reduction is more thermodynamically favorable, conditions more suitable for Fe(III)-reducers can also inhibit methanogenesis[46]. However, Fe(III) reduction consumes protons and leads to an increase in pH which can make conditions more favorable for methanogens[47]. Along the thaw gradient, an increase in pH and an increasing abundance of methanogens has been reported[24]. Acetotrophic methanogens can use Fe(III) reduction to maximize energy conservation from metabolism of acetate[48]. Shifts in $CH_4$ production pathway from $CO_2$ reduction to acetate cleavage along the thaw gradient was previously described[21,24]. At the same time, anaerobic oxidation of methane by methanotrophs can also be coupled to Fe(III) reduction. An increase in methane oxidation rates along the thaw gradient has been shown by Perryman et al.[26]. Our data clearly show that reactive Fe phases serve as an important and overlooked, terminal electron acceptor along the thaw gradient and thus could exert a significant control on net methane emissions.

Furthermore, the released aqueous $Fe^{2+}$ could be complexed by organic carbon[10] along the thaw gradient, inhibiting re-oxidation even when oxygen concentrations are high. The peatland at Stordalen mire could be a significant source of bioavailable iron to surrounding lakes and rivers, as has been shown at other permafrost environments[49]. It has also been shown that Fe(III) minerals can act as sieves for dissolved OM by selectively trapping terrestrially derived OM (enriched in aromatic moieties) on mineral surfaces at redox interfaces[10]. More work is needed to elucidate the chemical nature of associated organic carbon to determine its lability, but our data suggest that direct chelation or co-precipitation of Fe–C structures play an essential role in carbon protection.

In order to better predict future greenhouse gas emissions from thawing permafrost soils and improve the accuracy of existing climate models, it is therefore crucial to further determine Fe(III) reduction rates, its direct contribution to $CO_2$ emissions from peatland mires, and its competition with other microbial processes, such as e.g. methanogenesis or methanotrophy.

## Methods

**Site description and sample collection.** Stordalen Mire is a peatland 10 km southeast of Abisko in northern Sweden (68 22′ N, 19 03′ E)[50] which is underlain by quartz-feldspar-rich sedimentary rock (Geological Survey of Sweden). The study site is within the discontinuous permafrost region of northern Scandinavia and consists of three distinct sub-habitats which are common to northern wetlands: (1) a well-drained palsa underlain by permafrost, dominated by ericaceous and woody plants, (2) a bog with variable water table depth and some active thawing, dominated by *Sphagnum* spp. mosses, and (3) a fully thawed and inundated fen, dominated by sedges such as *Eriophorum angustifolium*[21,22]. Dissolved $O_2$

decreases with increasing water table depth in bog areas from $150 \pm 40\,\mu M$ dissolved $O_2$ at 10 cm depth to $20 \pm 10\,\mu M$ at 20 cm depth. In the fen, dissolved $O_2$ at 10 cm depth is $20 \pm 10\,\mu M$ and $10\,\mu M$ at 20 cm depth[26]. Growing season mean soil temperature at 10 cm depth below surface varies along the thaw gradient, from 6.2 $\pm 0.2\,°C$ in palsa to $7.2 \pm 0.3\,°C$ in bog, and to $7.6 \pm 0.3\,°C$ in the fen[51]. In this study, the three sub-habitats were ordered following a temporal succession of apparent time from palsa, to bog and fen, a space-for-time approach, as has been done before[21] following the classification of Johansson et al.[22]. This approach is limited in that (1) permafrost thaw progression through these three thaw stages is not necessarily linear[21], (2) intermediate thaw stages (e.g. collapsed palsa) are not accounted for, and (3) it does not fully capture the heterogeneity of the landscape. However, this approach provides the best available estimate of how palsa mires will evolve with progressive climate change and thus has been applied widely at this site. The palsa and bog areas are underlain by permafrost with a thickness of 10–20 m[52]. The active layer, depending on the surface topography, ranges from 0.5–1 m thickness at maximum thaw[52,53]. These three thaw stages cover ~98% of the mire's non-lake surface[21]. A thaw-dependent shift in these habitats was observed from 1970 to 2000 during which palsa regions collapsed and bog and fen areas increased by 17%[20]. At the same time, an increase in average annual temperature by 2.5 °C between 1913 and 2006 was measured, resulting in an annual mean temperature >0 °C during the recent decades[54]. The total precipitation also increased during this period of time to an annual average of 306 mm[54]. The expansion of wetlands after permafrost melt is a widespread characteristic of peatlands affected by permafrost thawing[55–58] and the successional shift from palsa to bog and fen areas has been documented in other northern peatlands[57–60].

In July 2018, cores were taken in duplicates along a gently collapsing thaw gradient from palsa to bog and fen (Supplementary Fig. 1). Stordalen mire is a protected area with other ongoing field research, thus the extent of coring is strictly limited due to the risk of accelerating permafrost thaw and/or disturbance to other long-term measurements, especially at sensitive sites like erosion fronts. However, extensive context data (https://polar.se/en/research-in-abisko/research-data/) from the Abisko scientific community are available, which ensures representative field sampling of a heterogeneous permafrost area, with cores taken following a transect along the direction of hydrological flow from palsa to bog and fen as described by Olefeldt and Roulet[28]. Given the restrictions in place, it was only possible to collect six cores per each thaw stage over three field campaigns (2017, 2018, and 2019) (Supplementary Figs. 1–5). A Humax corer of 50 cm length and 3-cm-diameter with inner core liners was used. The inner liners were washed three times with 80% ethanol, six times with sterile MilliQ water and sealed with sterilized butyl rubber stoppers until coring. Butyl rubber stoppers were boiled three times in deionized water and sterilized at 121 °C for 20 min in an autoclave. Sharp edges were cut into the end of the coring device to help cut the peat layer. A hammer was used to further sample the active layer. Hammering caused slight compaction of the cores, mainly in the organic horizon. Therefore, the recorded depths are not completely comparable to the initial soil profiles. Thus, the data are presented by the different soil layers (organic horizon, transition zone, and mineral horizon) and depth intervals are still reported to illustrate that the dataset represents the active layer depth along the thaw gradient. Stocks (mg/cm$^2$) were calculated considering compaction and compared to the content (mg/g) (Supplementary Fig. 9). In the palsa and bog area, cores were taken until the depth of the ground ice. Layers at the bottom of the core which contained predominantly ice were excluded from further analyses. Therefore, the soil investigated in this study represented the seasonally thawed active layer at Stordalen mire, ranging from 30 to 49 cm. The cores were stored vertically at 4 °C in the dark. Three cores representing desiccating palsa, bog, and fen were processed within 3–4 days (Supplementary Fig. 6). Due to detailed analysis of the first core set (Palsa A, Bog C, and Fen E), additional cores (Palsa B, Bog D, and Fen F) from each thaw stage were analyzed after storage for 7 months at 4 °C in the dark, which is not ideal, but still could be used to determine if preservation of the carbon by reactive iron was stable over longer time periods (Supplementary Figs. 1 and 3). The long-term stored core Palsa B still showed higher abundance of reactive iron-associated organic carbon than Bog D and Fen F, but less than Palsa A which could be due to natural variability, long-term storage, or because it was taken closer to the collapsing edge (Supplementary Figs. 1 and 3). In 2019, an additional three cores (1 per thaw stage) (Palsa a, Bog c, and Fen e) were collected and analyzed under same conditions and with the same methods as Palsa A, Bog C, and Fen E from 2018. The cores taken in 2018 and 2019 were compared to triplicate cores previously collected in September 2017 at each thaw stage with a Pürckhauer corer and processed directly after sampling, to show that the trends are representative for the whole mire (Supplementary Figs. 1, 4 and 5). The organic horizon of the bog and fen replicate cores could not be sampled with the Pürckhauer corer. The palsa cores captured the whole active layer profile. The replicate cores showed the same trends of 6 M HCl extractable iron. Readily extractable Fe (defined by 0.5 M HCl extractable iron) showed similar trends to the sodium dithionite citrate or hydroxylamine-HCl extraction for all three thaw stages (Supplementary Fig. 4). The same trend of TOC along the thaw gradient was observed (Supplementary Fig. 5).

**Porewater sampling and analysis**. The cores were kept in a vertical position during transfer into an anoxic glovebox (100% $N_2$). Three different sections were identified by texture and color changes: (1) an organic horizon on top, (2) a middle

transition zone between the organic-rich and mineral-rich layer, and (3) a mineral horizon at the bottom (Supplementary Fig. 1). Rhizon porewater samplers (Rhizosphere research products, Netherlands) with a porous sampling area of 10 cm and 0.15 μm pore size were used to extract porewater from three different depths, resulting in one sample representing each organic horizon, transition zone, and mineral horizon. The extracted porewater was analyzed for dissolved Fe (total and Fe(II)), organic carbon (DOC), and fatty acids. For total Fe and Fe(II), the supernatant was acidified in 1 M HCl and quantified spectrophotometrically in triplicate with the ferrozine assay[61]. DOC was quantified in triplicate with a TOC analyzer (High TOC II, Elementar, Elementar Analysensysteme GmbH, Germany). For the DOC analysis, the calibration curves were $r^2 > 0.999$ and the standard deviations of the triplicate analysis were <2%. Inorganic carbon was removed by acidifying the samples with 2 M HCl prior analysis. High performance liquid chromatography (HPLC; class VP with refractive index detector (RID) 10A and photo-diode array detector SPD-M10A VP detectors; Shimadzu, Japan) was used to determine the fatty acid concentrations.

**Core splitting**. The soil cores were removed from their liners under a $N_2$ atmosphere. Each core was sectioned into an organic horizon of varying thickness (4–10 cm), a transition zone (3–5 cm), and mineral horizon (4–10 cm) (Supplementary Fig. 1), following Ryden et al.[27]. The transition zone represents the boundary between organic and mineral horizon and was additionally defined due to distinct geochemical conditions in the porewater analysis in the middle of the active layer near the boundary between organic and mineral horizon. Calculated bulk densities as a function of soil OM following Bockheim et al.[62] were consistent with other studies conducted at Stordalen mire[27] (Palsa A: organic horizon: $0.03 \pm 0.01\,g/cm^3$, transition zone: $0.08 \pm 0.02\,g/cm^3$, mineral horizon: $0.84 \pm 0.26\,g/cm^3$; Bog C: organic horizon $0.08 \pm 0.01\,g/cm^3$, transition zone $1.29 \pm 0.04\,g/cm^3$, and mineral horizon $1.74 \pm 0.01\,g/cm^3$, Fen E: organic horizon $0.21 \pm 0.02\,g/cm^3$, transition zone $1.97 \pm 0.2\,g/cm^3$, and mineral horizon $1.72 \pm 0.01\,g/cm^3$). Sub-samples were homogenized and weighed into 10 mL glass vials and kept frozen at −20 °C prior to subsequent analysis.

**Selective extractions**. The soil layers were subjected to several chemical extractions to quantify the different iron phases. The soils were kept frozen prior to analysis, then dried at 20 °C under anoxic conditions until no further weight loss was observed (1 day). Prior to use, all glassware was washed with 1 M HCl for 10 min, flushed three times with deionized water and once with MilliQ water. Afterward glassware was sterilized at 180 °C in the oven for 4.5 h. 0.3 g of dry soil was weighed into a 10-mL glass vial with 6.25 mL extractant and $N_2$ headspace. Following the extraction, as detailed below, all samples were centrifuged at room temperature for 10 min at $5300 \times g$. After centrifugation, the supernatant was decanted into another 10 mL glass vial. Each extraction was performed in duplicates for each layer. Throughout the extraction, samples were kept in the dark under anoxic conditions ($N_2$ atmosphere). The extracts were analyzed for Fe and DOC as described above. Additionally, the samples were acidified in 1% (v/v) $HNO_3$ and analyzed in duplicates by microwave plasma atomic emission spectroscopy (MP-AES)/inductively coupled plasma mass spectrometry (ICP-MS). To get the total iron (Fe), phosphorous (P), and sulfur (S) concentrations, the extracts were analyzed using ICP-MS. To further obtain aluminum (Al) concentrations and a cross-check of the total Fe concentrations, the MP-AES was used for analysis (Supplementary Fig. 10). The illustrated iron values throughout the whole study represent the iron values obtained by the ferrozine assay (for differences in iron concentrations through the different analysis, see Supplementary Fig. 12). Due to dark color of the extracts which can disturb the spectrophotometric measurement during ferrozine complexation, the absorbance of blanks (sample diluted in 1 M HCl or hydroxylamine-HCl) was measured and later subtracted to avoid overestimation of iron concentrations. At the same time, samples and standards are heavily diluted in 1 M HCl and hydroxylamine-HCl (1:1000 and 1:10,000) before spectrophotometric quantification. Given these high levels of dilution, no matrix effects of the extractant solution remain. For the ferrozine assay, the calibration curves were $r^2 > 0.999$, and the standard deviations of the triplicate analyses were <1%. For the ICP-MS, the calibration curves were $r^2 > 0.999$, and the standard deviations of the triplicate analyses were <5%. For the MP-AES, the calibration curves were $r^2 > 0.993$ and the standard deviations of the triplicate analysis were <10%. The results of all Fe analysis (ferrozine assay, MP-AES/ICP-MS analysis) show all the same trends with depth and along the thaw gradient (Supplementary Fig. 12). For additional extractant-specific experimental parameters, see below.

*6 M HCl*. To extract more crystalline iron phases of the soil layers, such as poorly reactive sheet silicate Fe or FeS species, dried samples were subjected to 6 M HCl extraction at 70 °C for 24 h[63–65].

*Sodium pyrophosphate*. The sodium pyrophosphate extraction was performed following Coward et al.[8] at pH 10 to determine the colloidal or OM-chelated iron and bound organic carbon (Supplementary Figs. 7 and 8).

*Hydroxylamine-HCl.* To extract the short ranged ordered (SRO) Fe oxides, an acidic hydroxylamine-HCl (pH <2) extraction was carried out under the same conditions as the sodium pyrophosphate extraction[8].

*Dithionite–citrate.* Extractions were conducted using a solution of 0.27 M trisodium citrate, 0.11 M sodium bicarbonate, and 0.1 M sodium dithionite (total ionic strength: 1.85 M), as previously described[5] (see also Supplementary Method 1). This extraction was used to also quantify the reactive iron minerals but in particular the organic molecules binding to it (released during iron mineral dissolution). Instead of heating to 80 °C as described by Lalonde et al.[5], the dithionite–citrate extraction was performed under the same conditions as the sodium pyrophosphate and hydroxylamine-HCl extraction (on a rolling shaker at room temperature for 16 h) for better comparison between the different extractions and to prevent carbon alteration during heating (see also Supplementary Method 1, Supplementary Figs. 7 and 8), following previous studies[8,66,67]. The citrate addition as a metal ion complexing agent was necessary to avoid underestimation of iron and organic carbon as a result of complexation or mineral precipitation during extraction (Supplementary Method 1). Without citrate addition, we obtained 64 ± 3% less iron and 57 ± 28% less carbon after sodium dithionite reductive dissolution. As described in Lalonde et al.[5], we also used a 1.85 M sodium chloride/0.11 M sodium bicarbonate extraction as a control experiment under the same conditions (same solid:solution ratio, temperature, time, ionic strength) to distinguish between organic carbon which is readily desorbed and organic carbon which is released by the reduction of iron(III) minerals (Supplementary Figs. 7 and 8). To determine the DOC background concentrations caused by the trisodium citrate, blanks (trisodium citrate sodium bicarbonate solution) were analyzed during each measurement. The background concentration was later subtracted from the total DOC value, as well as the DOC concentration of the control experiment (sodium chloride sodium bicarbonate solution), resulting in the OC which is released by the reduction of reactive iron (Supplementary Table 1). Microbial activity during and after the dithionite–citrate extraction and trisodium citrate sodium bicarbonate extraction can be excluded due to high salt content (ionic strength of 1.85 M). Rath et al.[68] previously showed that salinity exerts a strong inhibitory effect on a range of microbial processes in soil and that acute toxic effects occurred immediately (within 2 h).

**TOC analysis.** To quantify the TOC (Fig. 4, Supplementary Table 1 and Supplementary Figs. 2, 3 and 5), soil samples from each layer were dried at 60 °C until the weight remained constant. The dry soils were then ground and acidified with 16% HCl to remove the inorganic carbon. After washing with deionized water and subsequent drying, the TOC content was analyzed by an Elementar vario El (Elementar Analysensysteme GmbH, Germany). The TOC content goes in line with previously reported values[69–71]. For the TOC analysis, the calibration curves were $r^2 > 0.998$ and the standard deviations of the triplicate analysis were <1%.

**EXAFS analysis.** Samples of the transition zone and the mineral horizon of the two-end members, palsa and fen, were dried under an $N_2$ atmosphere and stored anoxically in a glove box prior to analysis. Sample were then sealed in plastic multi-sample holders with Kapton polyimide tape and kept anoxic until they were transferred to a sample mount at the beamline. The sample holder was in a cryostat during analysis to limit beam damage and to prevent oxidation of Fe(II).

Fe K-edge X-ray absorption spectroscopic analyses were conducted at Beamline 11−2 at the Stanford Synchrotron Radiation Light source (SSRL) in Menlo Park, CA. The Si(220) phi = 0° monochromator was used, and beam size of 1 mm vertical and 10 mm horizontal. EXAFS fluorescence spectra were collected with the PIPS detector simultaneously with the transmission spectrum of Fe foil, which was used for internal energy calibrations. Multiple scans (3–4) per sample were acquired as necessary to achieve satisfactory data quality.

Scans were calibrated to 7112 eV (the first inflection point of Fe(0)), and then averaged over three or four scans using SixPack software. They were deglitched at 7250 and 7600 eV, and then normalized with the E0 value, determined by finding the inflection point of the first derivative of each sample. A set of Fe reference compounds was used to perform linear combination fitting (LCF) of EXAFS spectra in SixPack from chi values of 2–12 with an *x*-weight of 3. Non-negative fits were performed. Reference compounds were chosen based on prior knowledge of the sample including, for example, criteria such as elemental composition (determined by element composition of in the soil extracts), site characteristics (e.g. redox conditions, pH), and principal component analysis (PCA). PCA determines the number of distinct species in series of spectra. All contributions below 5 wt% were eliminated since we have previously determined that the limit of detection for mixed Fe species is around 5 wt%[29,72]. We determined the best least-square fitting based on fitting parameters such as the reduced chi$^2$ ($X^2$) and *R*-factor values. The best fits were reference samples such as natural nontronite and ferrosmectite (referred to as Fe clays) obtained from the Clay Mineral Society, Fe(II)-citrate and Fe(III)-citrate (referred to as Fe(II)-OM and Fe(III)-OM) which were prepared and analyzed as described in Daugherty et al.[73], mackinawite (referred to as FeS) which was prepared and analyzed as described in Troyer et al.[74], and 2-line ferrihydrite (referred to as poorly crystalline Fe), prepared and analyzed as described in Borch et al.[75] (Fig. 3).

**Correlative SEM and nanoSIMS.** The two end-members, palsa and fen, were analyzed using SEM and nanoSIMS (Fig. 4 and Supplementary Fig. 11) using only the free particles of the fine fraction of the organic horizon, the transition zone and the mineral horizon. As described by Kopittke et al.[76] and Keiluweit et al.[77], subsamples of each layer (1 mg) were dispersed in 10 mL of anoxic deionized water and gently shaken to obtain the free organo-mineral particles from the fine fraction of the soil. Then, 100 µL of the suspension was placed onto a silica wafer and dried under an $N_2$ atmosphere. The samples were sputter-coated with 12 nm platinum (Pt) using a Bal-Tec SCD005 sputter coater.

To characterize the organo-mineral particles of the fine fraction by size and crystallinity and identify representative particles, a field emission scanning electron microscope (FE-SEM; Jeol JSM-6500F), equipped with secondary electron detector, was used prior to nanoSIMS analysis. The acceleration voltage was set to 5 kV, with a working distance of 10 mm.

The nanoSIMS analysis were performed at the Cameca nanoSIMS 50L of the Chair of Soil Science (TU München, Germany). Prior to the measurements, the samples were additionally coated with Au/Pd layer (~30 nm) to avoid charging during the analysis. The $Cs^+$ primary ion beam was used with a primary ion impact energy of 16 keV. Prior to the final measurement, any potential contaminants and the Au/Pd coating layer were sputtered away at $50 \times 50$ µm with a high primary beam current (pre-sputtering). To enhance the secondary ion yields, $Cs^+$ ions were implanted into the sample during this pre-sputtering process. The primary beam (~1.2 pA) was focused at a lateral resolution ~100 nm and scanned over the sample with $^{12}C^-$, $^{16}O^-$, $^{12}C^{14}N^-$, $^{31}P^-$, $^{32}S^-$, $^{27}Al^{16}O^-$, and $^{56}Fe^{16}O^-$ secondary ions collected using electron multipliers. To compensate for any charging of the non-conductive mineral particles, the electron flood gun was used. All analyses were performed in imaging mode. For every layer, four representative spots were analyzed to obtain a reliable data basis for the spatial distribution of $^{12}C^-$ and $^{56}Fe^{16}O^-$. Ion images of $30 \times 30$ µm field of view, 30 planes with a dwell time of 1 ms/pixel, 256 pixels × 256 pixels were recorded. The estimated depth resolution with 16 keV $Cs^+$ ions was 10 nm.

Finally, the nanoSIMS images were analyzed using the Open MIMS Image plugin available within ImageJ (available free-of-charge at https://imagej.nih.gov/ij/. All presented images were corrected for the electron multiplier dead time (44 ns), as well as drift corrected, and the planes accumulated. A median filter was applied on all images.

**MPN counts.** A growth-dependent approach, MPN counts (Fig. 2), was performed on the soil samples from the different depths of the cores in seven replicates. This is a useful way to quantify Fe(III)-metabolizing bacteria as there is a lack of specificity in the potential genes used for Fe(III) reduction and has the strength of directly showing that the microorganisms are capable of reducing Fe(III). The three soil layers (organic horizon, transition zone, and mineral horizon) were homogenized and used for preparing a dilution series. Each tube of the dilution series contained 9 mL of anoxic freshwater media. Then, 5 mM sodium acetate, 5 mM sodium lactate, and 5 mM 2-line ferrihydrite (chemically synthesized as previously described[78]) were added as amendments to the anoxic media (0.6 g/L $KH_2PO_4$, 0.3 g/L $NH_4Cl$, 0.025 g/L $MgSO_4 \times 7H_2O$, 0.4 g/L $MgCl_2 \times 6H_2O$, $CaCl_2 \times 2H_2O$, 22 mM $NaHCO_3$, 1 mL/L trace element according to Widdel et al.[79], 1 mL/L vitamin solution after Widdel and Pfennig[80], and 1 mL/L selenite/tungstate solution according to Widdel[81]) before preparing the dilution series. The headspace in the dilution series was $N_2$:$CO_2$ (90:10). To the first tube of a dilution series, 1 g of soil was added, and a 10× dilution series up to a dilution of $10^{-12}$ was prepared, as has been previously done with sediments by Laufer et al.[82]. MPN enumerations of Fe(III)-metabolizing bacteria were set up in 96-well plates with seven replicates for each dilution[78,83]. To each well, 900 µL of the media was added, and then 100 µL of the respective dilution of the soil was added. Each dilution was inoculated into 7 wells, while 1 well served as a sterile control and remained uninoculated (no dilution of soil added, only anoxic media with lactate, acetate, and ferrihydrite amendments)[82]. For anoxic incubation, the Anaerocult system (Merck, Germany) was used, together with an $O_2$ indicator stick (Merck, Germany). Incubation was done for 8 weeks at 20 °C in the dark. Generally, anoxic MPN deep-well plates were evaluated visually for positive growth, as reduced-black Fe(II) minerals formed were easily detectable, meanwhile the control stayed rusty-orange[82]. To calculate the cell numbers (cells/g soil) from the positive MPN wells, all positive wells per dilution were counted and the MPN was calculated using the software program KLEE, applying confidence limits of Cornish and Fisher[84] and the bias correction after Salama[85,86]. Further isolation of Fe(III)-reducing bacteria was performed with same media and supplies via multiple round of dilution to extinction. DNA was extracted from the further isolated culture (after seven transfers) using the UltraClean® Microbial DNA Isolation Kit (MO BIO Laboratories, Carlsbad, CA, US). Then, 16S rRNA gene fragments were amplified using the 341F (CCTACGGGAGGCAGCAG) and 907R (CCGTCAATTCCTTTRAGTTT) primer pair and resulting amplicons were sent for Sanger sequencing (Eurofins GATC biotech, Konstanz, Germany). Sequence results were analyzed using nucleotide Basic Local Alignment Search Tool (BLAST) to identify the closest relative.

## Data availability

The data that support the findings of this study are included in a compressed Source Data file at https://doi.org/10.5281/zenodo.4106312. Other data are included in the Supplementary Information. Pre-processed data are available upon request.

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

## Acknowledgements

The authors would like to acknowledge the Abisko Research Station (Abisko, Sweden) for their support during sampling missions. We thank H. Miller (Colorado State University, Fort Collins, United States) for her assistance with EXAFS analysis, G. Harrington and J. Lugmeier (Munich, Germany) for nanoSIMS analysis and E. Stopelli (Zuerich, Switzerland) for ICP-MS measurements. This work was supported by the University of Tübingen (Programme for the Promotion of Junior Researchers grant to C.B.) and by the German Academic Scholar Foundation (scholarship to M.S.P.). Use of the Stanford Synchrotron Radiation Lightsource, SLAC National Accelerator Laboratory, is supported by the U.S. Department of Energy, Office of Science, Office of Basic Energy Sciences under Contract No. DE-AC02-76SF00515.

## Author contributions

The original hypothesis was formulated by C.B. and A.K. M.S.P., C.B., and A.K. designed the project, interpreted the data, and wrote the manuscript. M.S.P., C.B., and M.M. collected the samples. M.S.P. gathered the data presented in the main text. Supporting information from the 2017 campaign was collected by V.N., M.M., M.B., and C.B. T.B. conducted the synchrotron analysis and contributed to the data analysis and interpretation. C.H. and C.W.M., together with M.S.P., collected, analyzed, and interpreted the nanoSIMS data. J.M.B. and M.S.P. conducted the SEM analyses. T.S. contributed to project design and data interpretation. All authors contributed to the preparation of the manuscript.

## Competing interests

The authors declare no competing interests.
