## [Peer Review File · Nature Communications]

REVIEWER COMMENTS

Reviewer #1 (Remarks to the Author):

In this work the authors have used an elegant “space for time” approach to follow the dynamic interactions between iron and organic carbon across a permafrost thaw gradient; in which original dry and oxic tundra underlain by permafrost is progressively thawed giving rise to semi wet and reducing bog and eventually wet and even more reducing fen. They take depth core samples from each archetype space-time location and characterize the iron-carbon dynamics in three operationally defined layers, an organic layer nearest the surface, a transition zone, and a mineral layer at depth. The authors present a consistent set of data that indicates that iron-carbon associations are located primarily in the transition zone of the tundra, and that with space-time both the iron and carbon are mobilized as the wet and reducing conditions increasingly favor microbial Fe(III) reduction. They conclude that the “rusty carbon sink” will not protect carbon from remineralization in regions with thawing permafrost.

The discovery that iron-carbon associations might not offer protection from remineralization under sub-oxic or reducing conditions in and of itself is not novel (see recently published Chen et al., 2020 Nature Communications, and references therein, for a similar conclusion reached for humid (and thus wet and reducing) soils), *however*, as stated by the authors, there is very little knowledge of how the iron-carbon relationship plays out in high latitude permafrost peatlands, which are of prime importance in evaluating the contribution of soil carbon loss to climate change. *In this regard this work makes an important contribution to the knowledge base and is of interest to a wide research community.*

Overall I find the work to be solid and I agree with the interpretations, and due to the nature of the topic and the novelty of the findings in this important environment, I do feel

that the paper will influence thinking in this field. I have some general and some more specific concerns however, for the authors to address, detailed below.

Firstly, it is unclear how stratified the bog and particularly the fen environments are, in terms of their redox profiles. I wonder whether the surface layer is sufficiently aerated to provide a geochemical barrier to ultimate carbon loss to the atmosphere? In continental margin marine sediments for example, wet and reducing sediments at depth are overlain by oxic bottom water, such that there is an oxidized layer (mm's to cm's thick) at the sediment-water interface. This acts as an effective trap for many reduced species, as they diffuse upwards from the reducing zone, preventing their wholesale diffusion into seawater. Here iron oxyhydroxides are reprecipitated. By analogy might a similar process at the bog and fen surface recapture DOC such that DOC is effectively recycled between the evolving layers of the soil profile?

Secondly, I would like to see some tests and/or discussion about the effect of associated carbon on the stability of iron oxyhydroxides. The authors show that the isolate of Fe(III)-reducing bacteria is able to reduce ferrihydrite to Fe(II) (Fig. S10) but did they investigate whether and to what extent the isolate is able to reduce ferrihydrite organomineral? There is a significant body of evidence showing that organic carbon coatings and other iron-carbon interactions can stabilize ferrihydrite against mineral transformation and reductive dissolution (again see recently published Chen et al., 2020 Nature Communications, and references therein, where they discuss this in their Synthesis section), thus potentially limiting carbon release and/or remineralization.

Thirdly, whilst the authors do calculate the fraction of atmospheric carbon that the iron-bound carbon stock in permafrost soils represents, this amounts to 'only' 2-5%...so some discussion of the significance of the release of this stock for global climate change is warranted. The majority of carbon present in these soils appears to be unassociated with reactive iron and thus the importance of the "rusty carbon sink" in the context of the overall permafrost-carbon release-climate issue could be seen as rather minor.

Finally, how does the authors' "space for time" approach account for temperature changes inherent in the thaw process as a result of climate warming? It is unclear whether increased temperatures in the soil profiles will favor microbial activity such that DOC is oxidized before it ever has the chance to form iron-carbon associations, thus bypassing the "rusty carbon sink" entirely.

Some specific concerns below:

Did the authors characterize whether there are Fe and DOC fluxes into the system, from either lateral or top-down sources? It is coincident that the Fe and DOC increase along the thaw gradient, along with Fe(III)-reducing bacteria – but I wonder whether there are extrinsic sources of Fe or DOC that remain unquantified?

I have some queries regarding the selective extraction procedures. The authors do not appear to have calibrated their modified dithionate extraction method (room temperature for 16 hours) against the standard approach (80 degrees C for 15 minutes). Aqueous dithionate undergoes rapid degradation so extending their extractions to 16 hours may have reduced the 'reducing power' of the extractant. Whilst the results might be compared to the hydroxylamine-HCl extraction to potentially rule out any 'reducing power' problems (to determine whether both methods give similar results), the hydroxylamine-HCl extraction is performed at a lower pH and is thus more aggressive so these two extractions may not be directly comparable.

Reviewer #2 (Remarks to the Author):

Summary

Patzner et al. analyzed soil cores collected along a permafrost thaw gradient in northern Sweden to investigate the idea that the "rusty carbon sink" present in permafrost-underlain soils degrades as the permafrost thaws and the soils collapse. That is, Fe oxides present in well aerated palsas soils are gradually dissolved when the palsa collapses and floods, and the organic C associated with the Fe oxides is released and potentially accessible to microbial decomposition. This is an important concept that warrants further investigation – little is known with respect to how minerals stabilize organic C in these systems, and Fe oxides may be particularly important regulators of organic C (here and in other systems). The authors conclude that reactive Fe is lost along the thaw gradient with a concurrent loss of organic matter associated with the reactive Fe. Although this is a worthwhile topic with intriguing results, I do not think the work provides enough robust evidence to warrant publication in Nature Communications. Rather, they provide initial evidence that this could be an important process to consider, but its broader relevance to arctic systems given the limited dataset is not clear.

Major comments

The authors completed careful analytical work and provide good support for their chosen methods (e.g., comparing Fe from ferrozine vs AES/MS, correcting for controls). There are some places where more justification is needed, which is discussed in the comments below, but overall, their results are based on sound analysis.

A bigger concern is with the lack of environmental replication. Soils can be highly heterogeneous, and it is difficult to derive substantial conclusions from one core per area. The authors attempt to correct

for this by providing additional information on duplicate cores from the same areas and from additional cores from similar areas (analyzed with slightly different methods). Overall, this seems to be a fair comparison. The backup cores show similar trends (if not magnitudes) of reactive Fe. Trends for organic matter associated with reactive Fe are less supported since this was only available for the duplicate cores collected along the thaw gradient. Initially, I would have argued that organic C bound to reactive Fe seemed like a small and possibly negligible proportion of the total C, but I think their scaling up of its potential importance in the discussion justifies the study.

This leads to another major point of confusion that needs to be clarified. The methods describe the cores as being sectioned into organic, transition, and mineral horizons. It is also stated in the methods that core depths are not reported due to compaction of the core during collection. However, Figures 3, 4 and the corresponding SI Figures plot Fe and C concentrations as a function of specific depth intervals, with multiple intervals per horizon. In addition, the depths in Figures S7 and S8 appear to be wrong, as they start quite deep (~50 cm) and do not include an organic horizon.

If the point is that the rusty Fe sink degrades from the palsa to the bog to the fen, I would suggest averaging Fe and C concentrations (or proportions) by horizon across similar cores (the duplicate cores from 2018 as one set and the triplicate cores from 2017 as another set). That is, the data could look more similar to Figure 2b. Originally, I thought it was beneficial to see the depth trends, but it's not clear whether the reported depths are meaningful. Given the bulk densities, it may also be possible to calculate stocks (g/m²) in addition to concentrations for each horizon, which provides an even better estimate of how much Fe and C are lost. Ultimately quantification is quite difficult given that the soil collapse makes it challenging to compare horizons across the thaw gradient, but it may provide useful insight. Also, although reactive Fe concentrations do decrease, it is also apparent from the data that the increasing proportion of clays and/or other more crystalline minerals drives the change in the proportion of reactive Fe.

In another vein, I'm curious what the role of Fe(III)-organic complexes may be in protecting C. It appears that the palsa contains large proportions of this organic-bound Fe, and I wonder if the Fe has any stabilizing effect on the organic matter it's bound to. I don't know the answer to this, but it could be important to consider the transformation/loss of this phase in addition to Fe oxides given its prominence in the palsa soil.

Specific comments

- I. 11. What is the significance of the reactive Fe being concentrated in the transition zone? This is not discussed.
- I. 12. Suggest "During permafrost thaw..." to differentiate from annual freeze-thaw cycles.
- I. 48. Does 20 cm refer to depth in the soil or the thickness of the transition layer?
- I. 51. Dithionite will also extract crystalline phases like hematite (of which there probably isn't any in these soils), so what crystalline Fe phase does the HCl target? Fe in silicates? I think those would be at least partially dissolved in this extraction.
- I. 99. Please clarify how the assays used for Fe reducing bacteria were converted to MPN. I am not familiar with this analysis so it's not clear to me what in the 96-well plate was measured.
- I. 141. Agreed that soil collapse and removal of organic matter is a difficult consideration for understanding how Fe changes. Perhaps this is a place where calculating Fe stocks (g/m²) would be useful?
- I. 148. Suggest "...and on additional triplicate cores from each thaw stage analyzed with different but

comparable methods.”

Figure 3. Where does FeS fall in the extractions? HCl?

Figure 4. I’m surprised there’s such a remarkable decrease in organic C from the palsa to the bog and fen, especially given the increasingly water-logged conditions. Has this been examined? I would imagine from looking at these data that the total C stock decreases substantially.

I. 246. Suggest “However, this estimate does not account for deeper layers.”

I. 247. This statement that deeper layers could contain even more Fe-bound C is a bit of a leap, especially since Fe-bound C seems to be concentrated in the transition layer in these soils. Perhaps it could introduce more Fe into the system, but that isn’t really known.

I. 267. Yes, your EXAFS data support the increase in the proportion of organic-bound Fe(II)!

I. 335. Check this statement – it is difficult to tell from the graphs because of the different axes, but it appears to me that Palsa B has a higher peak reactive Fe concentration than Palsa A.

I. 353. Does samples in this sentence refer to pore waters? Why were the pore waters centrifuged if they were already filtered?

I. 376. The structure of this paragraph was a little confusing. I suggest revising as follows:
“Prior to use, all glassware was washed with 1 M HCl for 10 min, flushed three times with deionized water and once with MilliQ water. Afterwards glassware was sterilized at 180°C in the oven for 4.5 hours. 0.3 g dry soil was weighed into a 10 mL glass vial with 6.25 mL extractant and N₂ headspace. Following extraction, as detailed below, all samples were centrifuged at room temperature for 10 min at 5300 g.”

I. 391. I agree that the blank correction is important. Was the calibration curve also run in the matrix solution to check for interference?

I. 456. The EXAFS determinations were not clear to me. Were all reference spectra used in the linear combination fits? How did you decide which fit was the best? Were contributions rejected if they were below a certain value?

Table S1. Unclear what OC and what Fe is represented in the OC:Fe ratio.

Figure S8 (b and c). No organic horizon? Incorrect depths?

Reviewer #3 (Remarks to the Author):

Review for Patzner et al „Iron mineral dissolution releases iron and associated organic carbon during permafrost thaw “

Patzner et al present an interesting study on iron- and organic carbon cycling in thawing permafrost soils. Many earlier studies only focused on the release of “more stable” organic carbon (OC) from iron oxide dissolution (e.g. doi: 10.4319/lo.1998.43.6.1287). This study now confirms that iron-bound DOC is partly bioavailable and shows how it is released.

The topic of the study is timely and important in the context of climate change and biogeochemistry.

The many methods used to produce the collected data set are impressive. The results are interesting for readers from environmental microbiology, biogeochemistry, and to some extent for climate modelers. The study therefore merits publication.

The study is technically sound (with one exception) and can basically be published as is. I only have one major remark and very few marginal comments.

Major remark

One of the main hypothesis of this work is that thawing releases bioavailable DOC from iron oxides. However, as far as I can see, nothing was done to prevent the degradation of the bioavailable organic matter during the extraction experiments. The DOC concentrations released after dithionite extraction were corrected for DOC that is released by NaCl/NaHCO₃ of similar ionic strength. However, when shaking soil with water you create an ideal setting for microbial activity as you release large amounts of nutrients and OC while continuously shaking the mixture. Without defeating the undesired microbial consumption of OC you get up to 50 % less water-soluble OC per unit of soil (see figure 1a in doi: 10.1111/ejss.12256). This is a major problem of soil-water extracts, because the estimates of water-soluble organic carbon are underestimated without suppression of microbial activity. From the methods section it is not clear if microbial activity was suppressed during the experiments. If this was done and I missed it, please accept my apologies and nothing needs to be done. But if not, the estimates on water-soluble OC should be discussed.

The numbers for NaCl/NaHCO₃ control extraction are usually low (Fig. S2) so the error here should not be too large. Extraction with 6 M HCL should certainly lead to complete disinfection. But for the dithionite extraction I am uncertain. It appears that dithionite has only minor ecotoxicological effects, so microbial activity is probably not suppressed by the addition of dithionite.

Because the general conclusions of the study are not in danger, I think that a literature based discussion on this issue might be sufficient.

Minor remarks

Line 25-27: This is probably only true for humid climates. In arid regions iron oxides typically only change upon aging ("Ostwald ripening"), which happens on times-scales of years or even centuries.

Lines 140-143: This is also repeated in the introduction and the methods, which may not be necessary. Please consider shortening the manuscript here by removing repetition. .

Line 247-249: If this is true, than permafrost thawing is going to have a substantial impact from the release of iron-bound carbon alone. However, please bear in mind, that when a fen has evolved in the final stages of thawing it will act as a carbon sink again. This could be added here.

Line 362-364 and 392-401: Information on precision is already included in the figures with the error bars. But some information on accuracy or trueness would be great such as "Accuracy was better than xx %", if the data is available. If the NPOC mode for DOC analysis was used, how was inorganic carbon removed?

Line 393: Please spell out abbreviations (MP-AES/ICP-MS) and explain in a little more detail what was done. Which of the elements was measured by either of the methods? The information is given in the supplement text belonging to Fig. S3, but including it here in the methods section would be beneficial for the organization of the paper.

Fig. 2-4: it might be easier for readers to understand the numbers when the units are similar (either as mg/l or as mM)...

Fig. 4b: I assume that upper, middle and lower pictures are always organic, transition and mineral horizon. Maybe this could be stated in the caption or indicated in the figure.

Supplement text S3. Here the difference of the three methods used to measure iron is shown and discussed. Because reference material for such extraction done by the authors are not available it is difficult to obtain data on trueness and accuracy of each method. If you have no further information then each of the methods is equally well suited from an analytical point of view. In this case one may simply calculate the mean of all values. However, this is just a suggestion and should not be considered mandatory during revision.

In all an interesting study that I like to see published eventually.

Thank you for considering me as a reviewer,
Thomas Riedel

Response to Reviewers' Comments

We thank the three reviewers and the editor, Dr. Melissa Plail, for their careful evaluations of our manuscript titled "*Iron mineral dissolution releases iron and associated organic carbon during permafrost thaw*" (manuscript ID: NCOMMS-20-20682-T). We are grateful to the reviewers for their favorable review of our work and the constructive comments and suggestions that significantly improved the quality of this manuscript. We appreciate the opportunity to revise and resubmit our manuscript and have carefully taken every comment into consideration and revised our manuscript accordingly.

In this revision letter, the reviewers' comments are shown in italic font, our responses are in normal font and green. We highlighted the changes we made in the manuscript in blue in this letter. Please note that all line numbers refer to the track changes version of the revised manuscript (i.e., with all changes included). Below is the point by point response to the reviewers' comments.

Reviewer 1:

In this work the authors have used an elegant “space for time” approach to follow the dynamic interactions between iron and organic carbon across a permafrost thaw gradient; in which original dry and oxic palsa underlain by permafrost is progressively thawed giving rise to semi wet and reducing bog and eventually wet and even more reducing fen. They take depth core samples from each archetype space-time location and characterize the iron-carbon dynamics in three operationally defined layers, an organic layer nearest the surface, a transition zone, and a mineral layer at depth. The authors present a consistent set of data that indicates that iron-carbon associations are located primarily in the transition zone of the palsa, and that with space-time both the iron and carbon are mobilized as the wet and reducing conditions increasingly favor microbial Fe(III) reduction. They conclude that the “rusty carbon sink” will not protect carbon from remineralization in regions with thawing permafrost.

The discovery that iron-carbon associations might not offer protection from remineralization under sub-oxic or reducing conditions in and of itself is not novel (see recently published Chen et al., 2020 Nature Communications, and references therein, for a similar conclusion reached for humid (and thus wet and reducing) soils), however, as stated by the authors, there is very little knowledge of how the iron-carbon relationship plays out in high latitude permafrost peatlands, which are of prime importance in evaluating the contribution of soil carbon loss to climate change. In this regard this work makes an important contribution to the knowledge base and is of interest to a wide research community.

Overall, I find the work to be solid and I agree with the interpretations, and due to the nature of the topic and the novelty of the findings in this important environment, I do feel that the paper

will influence thinking in this field. I have some general and some more specific concerns however, for the authors to address, detailed below.

Author response: We would like to thank the reviewer for the comments on the “space for time” approach we have adopted to follow interactions between iron and organic carbon across a permafrost thaw gradient, and for highlighting the consistency and strength of our dataset, as well as for seeing the novelty of our findings for the understanding of rapidly changing permafrost environments.

The reviewer is very correct in making the link between our study and the recent study of Chen *et al.* (2020)¹ in Nature Communications. We were excited to see that this recent publication observed similar limitations in the ability of iron minerals to protect organic carbon, and thank the reviewer for recognizing that our study advances this concept to iron-carbon relationships in redox-dynamic, permafrost environments about which little is known. The study by Chen *et al.* (2020)¹ was not released when we submitted our study, but is very complimentary to this work and indeed highlights that “future studies should [...] assess the extent that the formation and destruction of Fe-cemented microaggregates contribute to OM persistence in redox-dynamic soils”. The Chen *et al.* (2020)¹ paper is therefore now cited in our manuscript and discussed in detail:

Page 3-4, line 35-40 (Introduction): “The inventory of reactive iron minerals in humid climates is highly dynamic as they precipitate and dissolve in response to changing redox conditions. Mineral soil slurry incubations previously showed organic matter protection by iron only under static oxic conditions (Chen *et al.*, 2020)¹. However, iron(III) mineral reduction and dissolution under oxygen limitation led to anaerobic mineralization of dissolved organic matter and soil organic matter by 74% and 32-41%, respectively (Chen *et al.*, 2020)¹.”

Page 17, line 302-307 (Discussion): “Chen *et al.* (2020)¹ showed the formation of more crystalline Fe phases under O₂ fluctuations and suggested a substantial loss of co-precipitated organic matter under repeated redox fluctuations with a remaining Fe-bound fraction of organic matter resistant to reductive dissolution. In the bog thaw stage, redox fluctuations may have led to such a “core Fe-MAOM structure” (Fe mineral associated organic matter) (Chen *et al.*, 2020¹) but constant reducing conditions in the fen caused a complete loss of Fe-bound carbon in the solid phase.”

Firstly, it is unclear how stratified the bog and particularly the fen environments are, in terms of their redox profiles. I wonder whether the surface layer is sufficiently aerated to provide a geochemical barrier to ultimate carbon loss to the atmosphere? In continental margin marine sediments for example, wet and reducing sediments at depth are overlain by oxic bottom water, such that there is an oxidized layer (mm's to cm's thick) at the sediment-water interface. This acts as an effective trap for many reduced species, as they diffuse upwards from the reducing zone, preventing their wholesale diffusion into seawater...here iron oxyhydroxides are reprecipitated...by analogy might a similar process at the bog and fen surface recapture DOC such that DOC is effectively recycled between the evolving layers of the soil profile?

Author response: We would like to thank the reviewer for making this connection between our system and the more well-studied marine systems. At our field site, the redox conditions are somewhat stratified with depth in the profile of the bog and fen, and it is a good suggestion to refer to this in the manuscript. Unfortunately, we did not measure redox conditions with depth ourselves, but these were recently reported for this site by Perryman *et al.* (2020)⁷. Dissolved O₂ decreases with increasing water table depth in bog areas from 150 ± 40 μM dissolved O₂ at 10 cm depth to 20 ± 10 μM at 20 cm depth. In the fen, dissolved O₂ at 10 cm depth is 20 ± 10 μM and 10 μM at 20 cm

depth. We now refer to Perryman *et al.* (2020)⁷ redox profiles in the text, as outlined below.

Page 19, line 345-349 (Materials and Methods): “Dissolved O₂ decreases with increasing water table depth in bog areas from 150 ± 40 μM dissolved O₂ at 10 cm depth to 20 ± 10 μM at 20 cm depth. In the fen, dissolved O₂ at 10 cm depth is 20 ± 10 μM and 10 μM at 20 cm depth (Perryman *et al.*, 2020⁷).”

Additionally, we hypothesize that reduced iron, Fe²⁺_{aq}, is moving upward from deeper layers after Fe(III) reduction and is subsequently oxidized to Fe(III) by O₂ closer to the surface in the bog, in a similar process to that described by the reviewer. We suggest that the upward movement of Fe²⁺_{aq} and oxidation by oxygen explains why we find reactive Fe in the organic bog and fen horizon (page 12, lines 203-207).

*Secondly, I would like to see some tests and/or discussion about the effect of associated carbon on the stability of iron oxyhydroxides. The authors show that the isolate of Fe(III)-reducing bacteria is able to reduce ferrihydrite to Fe(II) (Fig. S10) but did they investigate whether and to what extent the isolate is able to reduce ferrihydrite organomineral? There is a significant body of evidence showing that organic carbon coatings and other iron-carbon interactions can stabilize ferrihydrite against mineral transformation and reductive dissolution (again see recently published Chen *et al.*, 2020 Nature Communications, and references therein, where they discuss this in their Synthesis section), thus potentially limiting carbon release and/or remineralization.*

Author response: We agree with the reviewer’s comment that more information on the stabilizing effect of the carbon on the iron oxides themselves would be beneficial. This has been studied in the literature and, based on previous findings (Shimizu *et al.*, 2013)⁸, it is likely that Fe-associated organic carbon can either increase or decrease reduction

rates of iron(III) minerals, such as ferrihydrite, but not prevent it. As suggested, we have significantly expanded the discussion on the effect of associated organic carbon on the stability of iron oxyhydroxides in the text. We have also added additional information on the effect of Fe on OM stability (as was requested by reviewer 2):

Page 3-4, line 20-44 (Introduction): “The release of vast amounts of organic carbon during thawing of high-latitude permafrost is an emerging issue of global concern. Yet, the extent of greenhouse gas emissions from permafrost thaw remains unpredictable due to knowledge gaps related to controls on the fate of carbon in permafrost soils (Estop-Aragones *et al.*, 2018⁹). The mobility, lability and bioavailability of organic carbon is determined by a number of inter-connected physico-biogeochemical parameters and processes. One such parameter is the presence of reactive iron minerals (defined here as iron minerals that are reductively dissolved by the chemical reductant sodium dithionite, e.g. ferrihydrite or goethite), which are known to stabilize organic carbon by sorption and co-precipitation (Wagai & Mayer, 2007¹⁰; Kogel-Knabner *et al.*, 2008¹¹; Lalonde *et al.*, 2012¹²; Kleber *et al.*, 2015¹³). Fe-bound carbon can be protected by soil structural conditions (such as aggregate formation, macro-scale shifts in fluid flow paths), thus be less accessible to decomposer organisms (Asano & Wagai, 2014¹⁴; Totsche *et al.*, 2018¹⁵). At the same time, oxygen (O₂) diffusion is hindered further favoring soil organic matter preservation rather than decomposition (Asano & Wagai, 2014¹⁴; Totsche *et al.*, 2018¹⁵). Thus, Fe-C associations are thought to significantly influence long-term carbon storage in numerous environments (Kleber *et al.*, 2005¹³; Lalonde *et al.*, 2012¹²; Riedel *et al.*, 2013¹⁶; Coward *et al.*, 2017¹⁷).

Several studies already identified poorly crystalline Fe organic matter associations in the field or produced them in the lab, and demonstrated that they are resistant to microbial or chemical reduction (Henneberry *et al.*, 2012¹⁸; Eusterhues *et*

al., 2014¹⁹, Coward *et al.*, 2018²⁰). The inventory of reactive iron minerals in humid climates is highly dynamic as they precipitate and dissolve in response to changing redox conditions. Mineral soil slurry incubations previously showed organic matter protection by iron only under static oxic conditions (Chen *et al.*, 2020¹). However, iron(III) mineral reduction and dissolution under oxygen limitation led to anaerobic mineralization of dissolved organic matter and soil organic matter by 74% and 32-41%, respectively (Chen *et al.*, 2020¹). When mineral dissolution occurs, iron and carbon mobilization, increased carbon lability/bioavailability, and increased gaseous carbon loss as CO₂ and CH₄ follow (catalyzed by heterotrophic and methanogenic microorganisms) (Turetsky *et al.*, 2008²; Zona *et al.*, 2009³; Lipson *et al.*, 2010⁴; Olefeldt *et al.*, 2013⁶; Herndon *et al.*, 2015⁵). The extent to which the formation and dissolution of reactive Fe phases contribute to soil organic matter persistence in redox-dynamic permafrost soils remains unknown. “

Page 16-17, line 289-301 (Discussion): “As has been stated previously, reactive Fe phases require their own physicochemical protection in order to contribute to organic matter persistence (Chen *et al.*, 2020¹), and the presence of organic matter itself can both increase or decrease the stability of Fe(III) minerals depending on the geochemical conditions. High C/Fe ratios, shown to enhance the extent of bioreduction compared to pure reactive iron mineral phases (Shimizu *et al.*, 2013⁸), could be one explanation as to why the reactive Fe phases present in these soils were not resistant to mineral dissolution with permafrost thaw. High C/Fe ratios are thought to enhance reduction rates (Shimizu *et al.*, 2013⁸) due to the function of carbon as an electron shuttle (Roden *et al.*, 2010²¹), its role as a strong ligand for Fe complexation (Jones *et al.*, 2009²²), or its importance for particle aggregation (Amstaetter *et al.*, 2012²³). Additionally, co-association of aluminum and iron with organic matter could have further made this rusty

carbon sink more susceptible to reductive dissolution (Masue-Slowey *et al.*, 2011²⁴). On the other hand, reduction-resistant surface coatings, embedding in a composite or aggregate (Asano & Wagai, 2014¹⁴; Coward *et al.*, 2018²⁰), or higher Fe mineral crystallinity (Hall *et al.*; 2018²⁵) could prevent mineral dissolution.”

The stability of Fe-OM associations could be further investigated by examining the rates and extent to which the Fe(III)-reducing isolate reduces ferrihydrite organo-minerals as the reviewer suggests. However, these experiments would be extremely complicated. The outcome of such an experiment would have a strong bias depending on which organic carbon is used and by the Fe:C ratios (ThomasArrigo *et al.*, 2018²⁶; Shimizu *et al.*, 2013⁸). Thus, this requires not one experiment, but a whole suite of experiments using different variables. Ideally, *in-situ* organic carbon would be used for such experiments. Nevertheless, the question remains how to sterilize it to avoid the influence of other active microbes without changing the *in-situ* carbon speciation and associations. In summary, it is possible to do the kind of experiment the reviewer describes with the microbial isolate, however, the scale of work required would not be a modest undertaking and would be better suited to a dedicated follow-up study. Additionally, the papers we have discussed above already contribute significantly to our knowledge on the stabilization of iron minerals by organic carbon and provide relevant insight into whether Fe stabilization by carbon is likely in our environment. We have revised the text related to the experiments with the Fe-reducing isolate to reflect this:

Page 8, line 124-127 (Results): “The isolate was able to reduce ferrihydrite to Fe(II) whilst simultaneously consuming lactate and producing acetate (Fig. S12). It did not utilize acetate. Fe-associated organic carbon could further enhance or lower reduction rates of ferrihydrite (depending on the OC:Fe ratios), but is unlikely to prevent mineral dissolution (Shimizu *et al.*, 2013⁸).”

Thirdly, whilst the authors do calculate the fraction of atmospheric carbon that the iron-bound carbon stock in permafrost soils represents, this amounts to 'only' 2-5%...so some discussion of the significance of the release of this stock for global climate change is warranted. The majority of carbon present in these soils appears to be unassociated with reactive iron and thus the importance of the "rusty carbon sink" in the context of the overall permafrost-carbon release-climate issue could be seen as rather minor.

Author response: We thank the reviewer for the suggestion to broaden the discussion on the significance of the release of iron-bound carbon stock in permafrost soils for global climate change. Overall, the 7 to 20% Fe-bound carbon seems minor compared to the overall carbon stock. However, if this carbon is highly bioavailable, potentially more bioavailable than the un-associated organic carbon, it can have a huge effect, being equivalent to 2-5% of the amount of carbon currently present in the atmosphere, which is equivalent to between 2 and 5 times the amount of carbon released yearly through anthropogenic fossil fuel emissions. This is supported by recent studies that have shown that mineral associated organic carbon is more bioavailable than particulate organic carbon (Jilling *et al.*, 2018²⁷; Kleber *et al.*, 2015¹³; Williams *et al.*, 2018²⁸). Additionally, we also highlighted that not only the released organic carbon can directly lead to greenhouse gas emissions, but that also the iron(III) as a terminal electron acceptor and the released aqueous Fe²⁺ have the potential to decrease or increase greenhouse gas emissions because Fe(III) reduction is coupled to oxidation of organic carbon to CO₂. We therefore changed the text accordingly:

Page 16, line 277-280 (Discussion): "This Fe-bound carbon stock is equivalent to approximately 2-5% of the amount of carbon which is currently present in the atmosphere which is equivalent to between 2 and 5 times the amount of carbon released yearly through anthropogenic fossil fuel emissions."

Page 17, line 308-314 (Discussion): “It should be noted that it is not only the released carbon that can directly contribute to greenhouse gas emissions. The reduction of Fe(III) itself will also contribute to CO₂ emissions since it is directly coupled to the oxidation and mineralization of organic carbon. On the other hand, since Fe(III) reduction is more thermodynamically favorable, conditions more suitable for Fe(III)-reducers can also inhibit methanogenesis (Van Bodegom *et al.*, 2004²⁹). However, Fe(III) reduction consumes protons and leads to an increase in pH which can make conditions more favorable for methanogens (Wagner *et al.*, 2017³⁰).”

Finally, how does the authors’ “space for time” approach account for temperature changes inherent in the thaw process as a result of climate warming? It is unclear whether increased temperatures in the soil profiles will favor microbial activity such that DOC is oxidized before it ever has the chance to form iron-carbon associations, thus bypassing the “rusty carbon sink” entirely.

Author response: We thank the reviewer for this comment but have to admit that we are not 100% sure if we understood the question correctly. We have interpreted it that the reviewer would like to know if our “space for time” approach accounts for the fact that microbial respiration may be faster in warmer temperatures, and would like to know if this would lead to loss of carbon before it can be sequestered by the rusty carbon sink. There is continuous monitoring of soil temperature at Stordalen and we know that the growing season mean soil temperature at 10 cm depth below surface varies along the thaw gradient, from 6.2±0.2°C in palsa to 7.2±0.3°C in bog to 7.6±0.3 °C in the fen (Malhotra & Roulet, 2015³¹, Lupascu *et al.*, 2012³²). By sampling along the thaw gradient, we also capture the effect of warmer soil temperatures that will be associated with later thaw stages. The reviewer is correct that these warmer temperatures result in higher microbial activity. Hodgkins *et al.*, 2014³³, observed faster decomposition in the

fen and Woodcroft *et al.*, 2018³⁴ noted a significant increase in microbial cells per g soil along the thaw gradient. Therefore, we do expect that microbial activity increases along the thaw gradient, and this is an effect captured by our “space for time” approach. Mean annual Arctic air temperatures have increased at almost twice the rate of the global average during the past 100 years (IPCC, 2007³⁵). Future climate projections suggest that the mean annual air temperature will increase by 2.2. to 3.2°C in the 60 to 90°C latitudinal zone by 2050, so it is possible that the temperature gradient from palsa to fen could become more extreme in the coming years and this is not something captured by our approach.

However, we consider it to be unlikely that microbial activity would be so fast that the rusty carbon sink was bypassed because in the later thaw stages the “rusty carbon sink” is in fact destroyed by microbial Fe(III) mineral dissolution. “Bypassing” the rusty carbon sink would require evidence that reactive Fe(III) minerals exist but that all DOC has been oxidized. This is in contrast to what we actually observed in the fen, which is very high DOC concentrations and almost no reactive Fe(III).

We now have included the information on temperature changes along the thaw gradient in the manuscript.

Page 19, line 347-349 (Material and Methods): “Growing season mean soil temperature at 10 cm depth below surface varies along the thaw gradient, from 6.2 ± 0.2 °C in palsa to 7.2 ± 0.3 °C in bog, and to 7.6 ± 0.3 °C in the fen (Malhotra & Roulet, 2015³¹).”

Did the authors characterize whether there are Fe and DOC fluxes into the system, from either lateral or top-down sources? It is coincident that the Fe and DOC increase along the thaw gradient, along with Fe(III)-reducing bacteria – but I wonder whether there are extrinsic sources of Fe or DOC that remain unquantified?

Author response: We thank the reviewer for this important question. We carefully followed the hydrological characterization of Olefeldt & Roulet (2012)³⁶, describing the flow and transport of dissolved organic carbon in this permafrost peatland, when choosing our sampling points. Olefeldt & Roulet (2012)³⁶ carefully described the palsa to bog flow and catchments without any other extrinsic sources of Fe or DOC. The fen catchment is hydrologically influenced by palsa to bog flow and additionally by surrounding surface water bodies (ponds, river and lake). Our analysis of this surface water showed average DOC concentrations of 24.87 ± 6.68 mg/L and average Fe concentrations of 0.02 ± 0.02 mg/L. Thus, these sources cannot explain the high Fe and DOC concentrations measured in the fen. We have clarified this in the text accordingly.

Page 7, line 104-109 (Results): “The palsa to bog catchment was described previously to have no other extrinsic sources of Fe or DOC (Olefeldt & Roulet, 2012³⁶). The fen catchment however is hydrologically influenced by palsa to bog flow and by surrounding surface water bodies (ponds, river and lake) (Olefeldt & Roulet, 2012³⁶). Analysis of this surface water showed average DOC concentrations of 24.87 ± 6.68 mg/L and average aqueous Fe^{2+} concentrations of 0.02 ± 0.02 mg/L. Thus, these extrinsic surface water sources cannot explain the high Fe and DOC measured in the fen.”

I have some queries regarding the selective extraction procedures. The authors do not appear to have calibrated their modified dithionate extraction method (room temperature for 16 hours) against the standard approach (80 degrees C for 15 minutes). Aqueous dithionate undergoes rapid degradation so extending their extractions to 16 hours may have reduced the ‘reducing power’ of the extractant. Whilst the results might be compared to the hydroxylamine-HCl extraction to potentially rule out any ‘reducing power’ problems (to determine whether both

methods give similar results), the hydroxylamine-HCl extraction is performed at a lower pH and is thus more aggressive so these two extractions may not be directly comparable.

Author response: The reviewer is correct we didn't compare our dithionite extraction (room temperature for 16 hours) against the approach conducted at 80°C for 15 minutes. We deemed this approach to be inappropriate for our samples and scientific question and instead opted to use lower temperature but longer incubation time to ensure we would be able to characterize the Fe-bound carbon after extraction. The 80°C approach risks altering the carbon during the heating procedure, which could further influence the amount of extracted iron. Instead of adopting the 80°C approach, we carefully followed the protocol of Holmgren, 1967³⁷ and Loeppert and Inskeep, 1996³⁸. This approach has been widely used and validated in numerous studies e.g. Wagai & Mayer (2007)¹⁰, Wagai *et al.* (2013)³⁹, Coward *et al.* (2017)¹⁷. These studies applied a dithionite-citrate extraction with similar solvent molarities (0.1 M dithionite, 0.3 M citrate) at pH 7-8 for 16 hours on a shaker to target short-range ordered Fe(III) oxides and crystalline Fe(III) oxides. Our results are therefore comparable to these other studies which focus on similar organic-rich samples to our study, and which have validated the selectiveness of the extraction. Indeed, earlier studies have suggested that the 80°C approach, together with high citrate, contributes to more nonselective dissolution which is something we intended to avoid (Ryan & Gschwend, 1991)⁴⁰. Whilst not directly comparable to the 80°C approach, the selectiveness of our approach has been demonstrated in numerous studies and deemed to be reliable. We have now expanded on the choice of temperature and incubation time in the manuscript and revised the manuscript accordingly:

Page 25, line 489-494 (Material and Methods): “Instead of heating to 80°C as described by Lalonde *et al.* (2012)¹², the dithionite-citrate extraction was performed under the same conditions as the sodium pyrophosphate and hydroxylamine-HCl extraction (on a

rolling shaker at room temperature for 16 h) for better comparison between the different extractions and to prevent carbon alteration during heating (see also SI, S3), following previous studies (Holmgren, 1967³⁷; Loeppert & Inskeep, 1996³⁸, Coward *et al.*, 2017¹⁷).“

Page 6, line 64-72 (Supplementary Information):”

(2) *Temperature and incubation time*

Dithionite citrate bicarbonate extractions have been widely applied in various studies and were previously performed under two different temperatures and incubation time. One is conducted at room temperature at pH 7-8 for 16 hours on a shaker (Holmgren, 1967³⁷; Loeppert & Inskeep, 1996³⁸; Wagai & Mayer, 2007¹⁰; Wagai *et al.*, 2013³⁹, Coward *et al.*, 2017¹⁷) and the other conducted at 80°C for 15 minutes (Mehra & Jackson, 1958⁴¹; Lalonde *et al.*, 2012¹²). Due to the high organic carbon content of the soil samples, the standard approach at room temperature at neutral pH for 16 hours was chosen to avoid alteration of carbon during heating of sample to 80°C, which could further influence the amount of extracted iron. The approach conducted at 80°C is suspected to contribute more to nonselective dissolution (Ryan & Gschwend, 1991⁴⁰) in organic-rich samples.”

Reviewer 2:

Patzner et al. analyzed soil cores collected along a permafrost thaw gradient in northern Sweden to investigate the idea that the “rusty carbon sink” present in permafrost-underlain soils degrades as the permafrost thaws and the soils collapse. That is, Fe oxides present in well aerated palsa soils are gradually dissolved when the palsa collapses and floods, and the organic C associated with the Fe oxides is released and potentially accessible to microbial decomposition. This is an important concept that warrants further investigation – little is known with respect to how minerals stabilize organic C in these systems, and Fe oxides may be particularly important regulators of organic C (here and in other systems). The authors conclude that reactive Fe is lost along the thaw gradient with a concurrent loss of organic matter associated with the reactive Fe. Although this is a worthwhile topic with intriguing results, I do not think the work provides enough robust evidence to warrant publication in Nature Communications. Rather, they provide initial evidence that this could be an important process to consider, but its broader relevance to arctic systems given the limited dataset is not clear.

The authors completed careful analytical work and provide good support for their chosen methods (e.g., comparing Fe from ferrozine vs AES/MS, correcting for controls). There are some places where more justification is needed, which is discussed in the comments below, but overall, their results are based on sound analysis.

Author response: We would like to thank the reviewer for appreciating that our topic is worthwhile and provides intriguing results. We are not the first study to suggest that iron-organic carbon associations might be an important carbon sink in permafrost environments (Kleber et al., 2005⁴²; Mu et al., 2016⁴³; Pokrovsky et al., 2016⁴⁴; Herndon et al., 2017⁴⁵). However, to the best of our knowledge, changes in Fe-OC

stabilization during permafrost thaw has never been followed in detail and, as such, the extent to which stabilization of C by Fe minerals can be expected during permafrost thaw was unknown. We hope our study can fill this knowledge gap to some extent and encourage further investigations to deepen our understanding in iron-carbon cycling during permafrost thaw in other arctic environments. We have done our best to fully address the methodological concerns of the reviewer and feel that these changes substantially improve the manuscript. Details of the changes made are detailed below.

A bigger concern is with the lack of environmental replication. Soils can be highly heterogeneous, and it is difficult to derive substantial conclusions from one core per area. The authors attempt to correct for this by providing additional information on duplicate cores from the same areas and from additional cores from similar areas (analyzed with slightly different methods). Overall, this seems to be a fair comparison. The backup cores show similar trends (if not magnitudes) of reactive Fe. Trends for organic matter associated with reactive Fe are less supported since this was only available for the duplicate cores collected along the thaw gradient. Initially, I would have argued that organic C bound to reactive Fe seemed like a small and possibly negligible proportion of the total C, but I think their scaling up of its potential importance in the discussion justifies the study.

Author response: We thank the reviewer for this comment. We do see the issue with the number of cores sampled. Unfortunately, the extent of coring at this site is strictly limited due to the risk of accelerating permafrost thaw and/or disturbance of other long-term measurements. However, to further support the trends for organic matter associated with reactive Fe, we have been able to obtain and analyze three additional cores along the thaw gradient (one palsa core, one bog core, one fen core). These were sampled and analyzed using the same methods as the cores discussed in the main text but collected in July 2019. These were analyzed directly after collection which was the limitation that

we faced with our replicate cores from the 2018 dataset. The additional replicate cores, now included in the manuscript, strongly support our conclusions, consistently showing that reactive Fe-bound carbon is lost along the thaw gradient. This data can now be found in the SI (figure S7) and was referred to in the main text, see text accordingly:

Page 6, line 82-86 (Introduction): “**Fig. 1.** Field site Stordalen mire close to Abisko in the North of Sweden. The three main thaw stages are (1) palsa (marked in orange), (2) bog (in green) and (3) fen (in blue). The positions of the three cores analyzed in detail within 3-4 days of collection in 2018, which represent all three thaw stages, are shown in yellow. Additional cores (shown in white) were taken in 2018 and analyzed after 7 months of incubation at 4°C (S6, Fig. S6 and Fig. S8). Data for further replicates, taken in 2017 and 2019, is provided in the SI (S6, Figs. S6-S10).”

Page 11, line 173-180 (Results): “In the following, only the data from cores Palsa A, Bog C and Fen E are discussed (Fig. 1, Fig. 3 and Fig. 4), but observed trends are supported by further analyses conducted on cores analyzed in the same manner, but collected in 2019 (Palsa a, Bog c, Fen e) (S7), as well as cores collected at the same time as Palsa A, Bog C and Fen E in 2018 but stored for a longer period (Palsa B, Bog D, Fen F) (S8). We also observed similar trends in triplicate cores from each thaw stage, analyzed with different but comparable methods (S9). The collected cores capture the three thaw stages over three years (2017, 2018 and 2019) (S6) and show that the trends of Fe and organic carbon along the thaw gradient are robust. “

Page 21, line 396-401 (Material and Methods): “In 2019, an additional three cores (1 per thaw stage) (Palsa a, Bog c and Fen e) were collected and analyzed under same conditions and with the same methods as Palsa A, Bog C and Fen E from 2018. The cores taken in 2018 and 2019 were compared to triplicate cores previously collected in September 2017 at each thaw stage with a Pürckhauer corer and processed directly after

sampling, to show that the trends are representative for the whole mire (Fig. S6, Fig. S9 and Fig. S10).“

Page 14, line 173-176 (Supplementary Information):

“(2) Palsa a, Bog c and Fen e replicate cores in 2019 (1 core per thaw stage)

These cores were taken with a Humax corer in July 2019 (Figure S6, a, red). Also, this set of cores was immediately split and processed after sampling and thus is directly comparable to Palsa A, Bog C and Fen E.”

Page 16, line 193-198 (Supplementary Information):”

Figure S6. a, Position of cores taken along a thaw gradient at Stordalen mire (Abisko, Sweden). Yellow: Cores were immediately split and processed after sampling (3-4 days). White: Cores were stored at 4°C for 7 months and then processed. Red: Cores were immediately split and processed after sampling in the same manner as the cores in yellow.”

Page 17, line 210-217 (Supplementary Information): “

Figure S7. Extractions of replicate cores. a, Iron and b, Carbon concentration of cores Palsa a, Bog c and Fen e. Cores were taken in July 2019, split directly after sampling and immediately processed. The green box marks the organic horizon, grey box the transition zone and yellow box the mineral horizon. Errors indicate the range of duplicate analyses of each layer in each thaw stage. TOC was determined via combustion, whereas the carbon in the dithionite citrate and the control extract (sodium chloride bicarbonate) was determined with the carbon analyzer.”

This leads to another major point of confusion that needs to be clarified. The methods describe the cores as being sectioned into organic, transition, and mineral horizons. It is also stated in

the methods that core depths are not reported due to compaction of the core during collection. However, Figures 3, 4 and the corresponding SI Figures plot Fe and C concentrations as a function of specific depth intervals, with multiple intervals per horizon. In addition, the depths in Figures S7 and S8 appear to be wrong, as they start quite deep (~50 cm) and do not include an organic horizon.

Author response: We thank the reviewer for pointing out these issues. These are important considerations that we had discussed when deciding how to present the depth profiles, and we thank the reviewer for pushing us to present this data in the best way possible. We decided to report our results by sections (organic horizon, transition zone, mineral horizon) because a) this is an important control on the iron pool and b) the cores suffer from minor compaction making exact depth intervals unreliable. However, even though the depth intervals should be taken with some caution due to compaction during the coring itself, we believe that overall it is still meaningful for illustrating that the dataset is representing the active layer depth along the thaw gradient. The compaction is also of more concern in some depths than in others, e.g. the organic horizon has lower bulk density, is more porous and is therefore more easily compressed. The dense mineral horizons, on the other hand, are not so heavily compacted. Thus, for mineral horizons, the layer thickness can be considered reliable. For this reason, we would prefer to keep the depth intervals and at the same time refer to the sections. All of this information is meaningful, it is just important to us that we draw the reader's attention to any factors which should be interpreted with caution.

Further, we thank the reviewer for highlighting issues with the presentation of the replicate cores which we had not adequately explained in the first submission and with wrong background colours for the different soil layers (organic horizon/transition zone/mineral horizon). The replicate cores were taken with a different type of corer

(Pürckhauer corer) which has one open side, in contrast to the fully enclosed Humax corer used for the main cores. For the palsa cores, the whole depth profile was successfully captured. However, the organic horizon of the bog and fen cores was impossible to retrieve during coring and thus are not reported in Figure S9 and S10. As the wet organic horizon of bog and fen was not solid enough to be retained by the corer and could not be collected. In bog and fen, the transition zones and mineral horizons were successfully sampled, confirming the loss of reactive iron along the thaw gradient at several different spots in the permafrost peatland mire. We thank the reviewer again, apologize that the issue with the organic horizon was not well-addressed in the previous version and now changed the text accordingly:

Page 21, line 401-403 (Material and Methods): "...representative for the whole mire (Fig. S6, Fig. S9 and Fig. S10). The organic horizon of the bog and fen replicate cores could not be sampled with the Pürckhauer corer. The palsa cores captured the whole active layer profile. The replicate cores showed..."

Page 14-15, line 182-188 (Supplementary Information):

"(4) Triplicate cores in each thaw stage (3 cores per thaw stage)

Triplicate cores in each thaw stage were taken with a Pürckhauer corer in September 2017 and immediately processed after sampling (Figure S6, a, orange, green and blue cores). The whole depth profile was successfully captured for all palsa triplicate cores. The organic horizon was lost for the bog and the fen triplicate cores during coring, thus are not reported in Figure S9 and Figure S10. In bog and fen, the transition zones and mineral horizons were successfully sampled, confirming the loss of reactive iron along the thaw gradient at several spots in the permafrost peatland mire. "

Page 19, line 225-233 (Supplementary Information): "

Figure S9. Iron extractions of replicate cores taken with a Pürckhauer corer: a, Palsa (68°21'26.56"N, 19° 3'0.19"E), b, Bog (68°21'16.02"N, 19° 2'49.21"E), c, Fen (68°21'17.16"N, 19° 2'36.29"E). Each core was divided into layers in the field and immediately processed. The organic horizon for the triplicate cores of bog and fen were lost during sampling and thus, are not reported in this figure. All replicates represent the active layer in September 2017. The bog and the fen soils were waterlogged. The green box marks the organic horizon, grey box the transition zone and yellow box the mineral horizon. Errors indicate the range of duplicate analyses of each layer in each thaw stage.”

Figure S10. TOC of replicate cores taken with a Pürckhauer corer: a, Palsa (68°21'26.56"N, 19° 3'0.19"E), b, Bog (68°21'16.02"N, 19° 2'49.21"E) and c, Fen (68°21'17.16"N, 19° 2'36.29"E). Each core was divided into layers in the field and immediately processed. The organic horizon for the bog and fen triplicate cores were lost during sampling and thus are not reported in this figure. All replicates represent the active layer in September 2017. The bog and fen soils were waterlogged. The green box marks the organic horizon, grey box the transition zone and yellow box the mineral horizon. TOC was determined via combustion. Errors indicate the range of duplicate analyses of each layer in each thaw stage.”

If the point is that the rusty Fe sink degrades from the palsa to the bog to the fen, I would suggest averaging Fe and C concentrations (or proportions) by horizon across similar cores (the duplicate cores from 2018 as one set and the triplicate cores from 2017 as another set). That is, the data could look more similar to Figure 2b. Originally, I thought it was beneficial to see the depth trends, but it's not clear whether the reported depths are meaningful. Given the bulk densities, it may also be possible to calculate stocks (g/m^2) in addition to concentrations for each horizon, which provides an even better estimate of how much Fe and C are lost. Ultimately quantification is quite difficult given that the soil collapse makes it challenging to compare horizons across the thaw gradient, but it may provide useful insight. Also, although reactive Fe concentrations do decrease, it is also apparent from the data that the increasing proportion of clays and/or other more crystalline minerals drives the change in the proportion of reactive Fe.

Author response: We thank the reviewer for giving such helpful feedback. We do see the advantage of averaging Fe and C concentrations by horizons across similar cores. We could average all triplicate cores from 2017. However, as the reviewer also pointed out earlier, the methods applied slightly vary between the duplicate cores from 2018. Thus, we hesitate to average the Fe and C concentrations over sets of different cores. As mentioned above, we would like to keep the depth intervals as they do have meaning but just should be interpreted with some caution.

We also appreciate the reviewer's suggestion to consider stocks, in addition to the content data we included already. Therefore, we now included the content (mg/g) versus stocks (mg/cm^2). In this case we have calculated the stocks both with compaction considered and without, averaged over horizons, in the Supplementary Information (see SI, Figure S1; see also following changes). In one plot, we calculated the stocks assuming that no compaction occurred during coring (SI, Figure S1, c) and in the other

we estimated compaction by comparing the final core length (as reported by the depth intervals in the main text) to the actual core hole (measured directly after coring) (SI, Figure S1, b). Based on the bulk densities, compaction was assumed for the palsa organic horizon ($0.03 \pm 0.01 \text{ g/cm}^3$) and palsa transition zone ($0.08 \pm 0.02 \text{ g/cm}^3$) and not for the dense palsa mineral horizon ($0.84 \pm 0.26 \text{ g/cm}^3$). Also, for the bog organic horizon ($0.08 \pm 0.01 \text{ g/cm}^3$) and for fen organic horizon ($0.21 \pm 0.02 \text{ g/cm}^3$) compaction was assumed, but not for the dense horizons (bog transition zone: $1.29 \pm 0.04 \text{ g/cm}^3$, bog mineral horizon: $1.74 \pm 0.01 \text{ g/cm}^3$, fen transition zone: $1.97 \pm 0.2 \text{ g/cm}^3$, fen mineral horizon: $1.72 \pm 0.01 \text{ g/cm}^3$).

Thus, where no compaction was assumed, the stock was calculated without compaction consideration. This means layer thickness used for the calculation is as reported by depth intervals in the main text. The stock is calculated using: bulk density (g/cm^3) * content (mg/g) * actual layer thickness in core (cm).

For the horizons where compaction was considered, a compaction factor was calculated using the difference between the actual core depth (as reported in the depth) and the core hole. The compaction factor for the palsa organic horizon and palsa transition zone was 2.31, for the bog organic horizon it was 5.00 and for the fen organic horizon it was 8.25. The compaction corrected stock was calculated as follows: compaction factor * bulk density (g/cm^3) * content (mg/g) * actual layer thickness in core (cm).

We think that this allows some indication of stocks whilst being transparent about any potential issues that the compaction may cause. However, as discussed above, we have chosen to continue discussing the absolute values of reactive Fe and reactive Fe-associated organic carbon in the main text with depth intervals.

We have now included this in the Supplementary Information and referred to in the main manuscript as follows:

Page 10-11, line 171-173 (Results): “Stocks (mg/cm^2) were calculated, but due to slight compaction, mainly for the organic horizons, not reported in the main text (S2).”

Page 20, line 383-384 (Material and Methods): “Stocks (mg/cm^2) were calculated considering compaction and compared to the content (mg/g) (S2).”

Page 3-5, line 13-49 (Supplementary Information):

”S2. Calculated stocks (mg/cm^2) versus content (mg/g) of reactive Fe and reactive Fe-associated organic carbon

During coring compaction occurred. Based on the bulk densities, compaction was assumed to have occurred in the palsa organic horizon ($0.03\pm 0.01 \text{ g}/\text{cm}^3$) and palsa transition zone ($0.08\pm 0.02 \text{ g}/\text{cm}^3$), but not for the dense palsa mineral horizon ($0.84\pm 0.26 \text{ g}/\text{cm}^3$). For the bog organic horizon ($0.08\pm 0.01 \text{ g}/\text{cm}^3$) and for fen organic horizon ($0.21\pm 0.02 \text{ g}/\text{cm}^3$), compaction was assumed but we assume that no compaction occurred in the dense horizons (bog transition zone: $1.29\pm 0.04 \text{ g}/\text{cm}^3$, bog mineral horizon: $1.74\pm 0.01 \text{ g}/\text{cm}^3$, fen transition zone: $1.97\pm 0.2 \text{ g}/\text{cm}^3$, fen mineral horizon: $1.72\pm 0.01 \text{ g}/\text{cm}^3$).

Thus, where no compaction was assumed, the stock was calculated without compaction consideration i.e. layer thickness used for the stock calculation is as reported by depth intervals in the main text:

$$\text{Stock} = \text{bulk density} * \text{content} * \text{layer thickness}$$

with stock in mg/cm^2 , bulk density in g cm^{-3} , content in mg g^{-1} and actual layer thickness in the core in cm.

For the horizons where compaction was assumed, a compaction factor was calculated from the difference between the actual core depth (as reported in the text) and the core hole (compaction factor for palsa organic horizon and palsa transition zone 2.31, for bog organic horizon 5.00 and for fen organic horizon 8.25). The compaction-corrected stock was calculated as follows:

$$\text{Stock} = \text{compaction factor} * \text{bulk density} * \text{content} * \text{layer thickness}$$

with stock in mg/cm^2 , bulk density in g cm^{-3} , content in mg g^{-1} and actual layer thickness in the core in cm.

Figure S1. Calculated stocks (mg/cm^2) versus absolute amounts (mg/g) of reactive Fe and reactive Fe-associated OC. a, Averaged absolute amounts of reactive Fe and associated organic carbon in mg per g soil per horizon (green: organic horizon, grey: transition zone, yellow: mineral horizon) for palsa, bog and fen. Error bars represent a combined standard deviation of the absolute values per depth (see Table S1) of duplicate extractions per horizons

of Palsa A, Bog C and Fen E. b, Calculated average stock of reactive Fe and reactive Fe-associated OC in mg per cm², compaction corrected. Green represents the organic horizon, grey the transition zone and yellow the mineral horizon in Palsa E, Bog C and Fen E. Error bars represent a combined standard deviation of the absolute values, bulk density and compaction per horizons of Palsa A, Bog C and Fen E. c, Calculated average stock of reactive Fe and reactive Fe-associated OC in mg per cm² without assuming compaction. The reported depth intervals in the main text were used to calculate the stock. Green represents the organic horizon, grey the transition zone and yellow the mineral horizon in Palsa E, Bog C and Fen E. Error bars represent a combined standard deviation of the absolute values and the bulk density per horizons of Palsa A, Bog C and Fen E.”

In another vein, I'm curious what the role of Fe(III)-organic complexes may be in protecting C. It appears that the palsa contains large proportions of this organic-bound Fe, and I wonder if the Fe has any stabilizing effect on the organic matter it's bound to. I don't know the answer to this, but it could be important to consider the transformation/loss of this phase in addition to Fe oxides given its prominence in the palsa soil.

Author response: We totally agree with the reviewer that complexation might be one preservation mechanism. This is also indicated by the OC:Fe ratios we discuss in the text (page 14-15, line 240-248, Results). Our study showed high reactive iron and associated organic carbon in palsa transition zones, and the loss of this reactive iron and associated organic carbon along the thaw gradient. The logical next step would be to further characterize the iron organic carbon associations present in the palsa soils and how stable they are before permafrost thaw is occurring. Of course, Fe(III)-organic complexes could also preserve organic carbon further along the thaw gradient. The released Fe²⁺ could complex with organic carbon and/or become oxidized to Fe³⁺ in the water column, forming Fe(III)-organic complexes. We now discuss the stabilization effect of iron organic matter associations in more detail as follows:

Page 3-4, line 20-44 (Introduction): “The release of vast amounts of organic carbon during thawing of high-latitude permafrost is an emerging issue of global concern. Yet,

the extent of greenhouse gas emissions from permafrost thaw remains unpredictable due to knowledge gaps related to controls on the fate of carbon in permafrost soil (Estop-Aragones *et al.*, 2018⁹). The mobility, lability and bioavailability of organic carbon is determined by a number of inter-connected physico-biogeochemical parameters and processes. One such parameter is the presence of reactive iron minerals (defined here as iron minerals that are reductively dissolved by the chemical reductant sodium dithionite, e.g. ferrihydrite or goethite), which are known to stabilize organic carbon by sorption and co-precipitation (Wagai & Mayer, 2007¹⁰; Kogel-Knabner *et al.*, 2008¹¹; Lalonde *et al.*, 2012¹²; Kleber *et al.*, 2015¹³). Fe-bound carbon can be protected by soil structural conditions (such as aggregate formation, macro-scale shifts in fluid flow paths), thus be less accessible to decomposer organisms (Asano & Wagai, 2014¹⁴; Totsche *et al.*, 2018¹⁵). At the same time, oxygen (O₂) diffusion is hindered further favoring soil organic matter preservation rather than decomposition (Asano & Wagai, 2014¹⁴; Totsche *et al.*, 2018¹⁵). Thus, Fe-C associations are thought to significantly influence long-term carbon storage in numerous environments (Kleber *et al.*, 2005¹³; Lalonde *et al.*, 2012¹²; Riedel *et al.*, 2013¹⁶; Coward *et al.*, 2017¹⁷).

Several studies already identified poorly crystalline Fe organic matter associations in the field or produced them in the lab, and demonstrated that they are resistant to microbial or chemical reduction (Henneberry *et al.*, 2012¹⁸; Eusterhues *et al.*, 2014¹⁹, Coward *et al.*, 2018²⁰). The inventory of reactive iron minerals in humid climates is highly dynamic as they precipitate and dissolve in response to changing redox conditions. Mineral soil slurry incubations previously showed organic matter protection by iron only under static oxic conditions (Chen *et al.*, 2020¹). However, iron(III) mineral reduction and dissolution under oxygen limitation led to anaerobic mineralization of dissolved organic matter and soil organic matter by 74% and 32-41%,

respectively (Chen *et al.*, 2020¹). When mineral dissolution occurs, iron and carbon mobilization, increased carbon lability/bioavailability, and increased gaseous carbon loss as CO₂ and CH₄ follow (catalyzed by heterotrophic and methanogenic microorganisms) (Turetsky *et al.*, 2008²; Zona *et al.*, 2009³; Lipson *et al.*, 2010⁴; Olefeldt *et al.*, 2013⁶; Herndon *et al.*, 2015⁵). The extent to which the formation and dissolution of reactive Fe phases contribute to soil organic matter persistence in redox-dynamic soils remains unknown.“

Page 16-17, line 289-307 (Discussion): “As has been stated previously, reactive Fe phases require their own physicochemical protection in order to contribute to organic matter persistence (Chen *et al.*, 2020¹), and the presence of organic matter itself can both increase or decrease the stability of Fe(III) minerals depending on the geochemical conditions. High C/Fe ratios, shown to enhance the extent of bioreduction compared to pure reactive iron mineral phases (Shimizu *et al.*, 2013⁸), could be one explanation as to why the reactive Fe phases present in these soils were not resistant to mineral dissolution with permafrost thaw. High C/Fe ratios are thought to enhance reduction rates (Shimizu *et al.*, 2013⁸) due to the function of carbon as an electron shuttle (Roden *et al.*, 2010²¹), its role as a strong ligand for Fe complexation (Jones *et al.*, 2009²²), or its importance for particle aggregation (Amstaetter *et al.*, 2012²³). Additionally, co-association of aluminum and iron with organic matter could have further made this rusty carbon sink more susceptible to reductive dissolution (Masue-Slowey *et al.*, 2011²⁴). On the other hand, reduction-resistant surface coatings, embedding in a composite or aggregate (Asano &Wagai, 2014¹⁴; Coward *et al.*, 2018²⁰), or higher Fe mineral crystallinity (Hall *et al.*; 2018²⁵) could prevent mineral dissolution.

Chen *et al.* (2020)¹ showed the formation of more crystalline Fe phases under O₂ fluctuations and suggested a substantial loss of co-precipitated OM under repeated redox

fluctuations with a remaining Fe-bound fraction of OM resistant to reductive dissolution. In the bog thaw stage, redox fluctuations may have led to such a “core Fe-MAOM structure” (Chen *et al.*, 2020¹) (Fe mineral associated organic matter) but constant reducing conditions in the fen caused a complete loss of Fe-bound carbon in the solid phase.”

Specific comments:

l. 11. What is the significance of the reactive Fe being concentrated in the transition zone? This is not discussed.

Author response: We thank the reviewer for making this clearer. We now discussed the significance of the transition zone in the abstract and discussion section in more detail. The text was changed accordingly:

Page 2, line 9-12 (Abstract): “We show through bulk (selective extractions, EXAFS) and nanoscale analysis (correlative SEM and nanoSIMS) that organic carbon is bound to reactive Fe primarily in the transition between organic and mineral horizons in palsa underlain by intact permafrost (41.8 ± 10.8 mg carbon per g soil, 9.9 to 14.8% of total soil organic carbon).”

Page 16, line 277-280 (Discussion): “This Fe-bound carbon stock is equivalent to approximately 2-5% of the amount of carbon which is currently present in the atmosphere which is equivalent to between 2 and 5 times the amount of carbon released yearly through anthropogenic fossil fuel emissions.”

Page 17, line 308-314 (Discussion): “It should be kept in mind that it is not only the released carbon that can directly contribute to greenhouse gas emissions. The reduction of Fe(III) itself will also contribute to CO₂ emissions since it is directly coupled to the

oxidation and mineralization of organic carbon. On the other hand, since Fe(III) reduction is more thermodynamically favorable, conditions more suitable for Fe(III)-reducers can also inhibit methanogenesis (Van Bodegom *et al.*, 2004²⁹). However, Fe(III) reduction consumes protons and leads to an increase in pH which can make conditions more favorable for methanogens (Wagner *et al.*, 2017³⁰).”

l. 12. Suggest “During permafrost thaw...” to differentiate from annual freeze-thaw cycles.

Author response: Agreed. Text is changed accordingly to “during permafrost thaw” (page 2, line 12, Abstract).

l. 48. Does 20 cm refer to depth in the soil or the thickness of the transition layer?

Author response: The 20 cm refers to the thickness of the transition layer. The text is changed to “20 cm layer thickness” (page 4, line 64, Introduction).

l. 51. Dithionite will also extract crystalline phases like hematite (of which there probably isn't any in these soils), so what crystalline Fe phase does the HCl target? Fe in silicates? I think those would be at least partially dissolved in this extraction.

Author response: We agree with the reviewer's comment. 6 M HCl would also extract some sheet silicate Fe (Poulton & Canfield, 2004)⁴⁶. Based on one of the following comments, we also added FeS targeted by 6 M HCl extraction (Heron *et al.*, 1994)⁴⁷ here. We changed the text accordingly:

Page 5, line 67-68 (Introduction): “and OM-chelated iron (sodium pyrophosphate), and more crystalline iron phases such as poorly reactive sheet silicate Fe or FeS species (6 M HCl).”

Page 24, line 477-478 (Material and Methods): “To extract more crystalline iron phases of the soil layers, such as poorly reactive sheet silicate Fe or FeS species, dried samples were subjected to a 6 M HCl extraction at 70°C for 24 h (Aller *et al.*, 1986⁴⁸; Heron *et al.*, 1994⁴⁷; Poulton & Canfield, 2005⁴⁶).”

l. 99. Please clarify how the assays used for Fe reducing bacteria were converted to MPN. I am not familiar with this analysis so it's not clear to me what in the 96-well plate was measured.

Author response: We apologize that this is not clearer in the text. We therefore changed it accordingly:

Page 29-30, line 581-607 (Material and Methods): “This is a useful way to quantify Fe(III)-metabolizing bacteria as there is a lack of specificity in the potential genes used for Fe(III) reduction and has the strength of directly showing that the microorganisms are capable of reducing Fe(III). The three soil layers (organic horizon, transition zone and mineral horizon) were homogenized and used for preparing a dilution series. Each tube of the dilution series contained 9 ml of anoxic freshwater media (0.6 g/L KH₂PO₄, 0.3 g/L NH₄Cl, 0.025 g/L MgSO₄ x 7 H₂O, 0.4 g/L MgCl₂ x 6 H₂O, CaCl₂ x 2 H₂O, 22 mM NaHCO₃, 1 mL/L trace element according to Widdel *et al.* (1983)⁴⁹, 1 mL/L vitamin solution after Widdel & Pfennig (1981)⁵⁰ and 1 mL/L selenite/tungstate solution according to Widdel (1980)⁵¹. 5 mM sodium acetate, 5 mM sodium lactate and 5 mM 2-line ferrihydrite (chemically synthesized as previously described (Straub *et al.*, 2005⁵²) were added as amendments to the anoxic media before preparing the dilution series. The headspace in the dilution series was N₂:CO₂ (90:10). To the first tube of a dilution series, 1 g of soil was added, and a 10 x dilution series up to a dilution of 10⁻¹² was prepared, as has been previously done with sediments by Laufer *et al.* (2016)⁵³. MPN enumerations of Fe(III)-metabolizing bacteria were set up in 96-well plates with

7 replicates for each dilution (Laufer *et al.*, 2016)⁵³. To each well, 900 µl of the media was added, and then 100 µl of the respective dilution of the soil was added. Each dilution was inoculated into 7 wells, while 1 well served as a sterile control and remained uninoculated (no dilution of soil added, only anoxic media with lactate, acetate and ferrihydrite amendments) (Laufer *et al.*, 2016⁵³). For anoxic incubation, the Anaerocult system (Merck, Germany) was used, together with an O₂ indicator stick (Merck, Germany). Incubation was done for 8 weeks at 20°C in the dark. Generally, anoxic MPN deep-well plates were evaluated visually for positive growth (Laufer *et al.*, 2016⁵³), as reduced-black Fe(II) minerals formed were easily detectable, meanwhile the control stayed rusty-orange. To calculate the cell numbers (cells/g soil) from the positive MPN wells, all positive wells per dilution were counted and the most probable number was calculated using the software program KLEE, applying confidence limits of Cornish & Fisher (1938)⁵⁴ and the bias correction after Salama (1978)⁵⁵.“

l. 141. Agreed that soil collapse and removal of organic matter is a difficult consideration for understanding how Fe changes. Perhaps this is a place where calculating Fe stocks (g/m²) would be useful?

Author response: As mentioned above, we agree that calculating the Fe stocks could be useful. As described above, we therefore now included calculated stocks (see SI, S2, page 3-5, line 13-49).

l. 148. Suggest “...and on additional triplicate cores from each thaw stage analyzed with different but comparable methods.”

Author response: We changed the text as suggested by the reviewer to “triplicate cores from each thaw stage, collected in 2017 and analyzed with different but comparable methods (S9).” (Introduction, page 11, lines 177-178)

Figure 3. Where does FeS fall in the extractions? HCl?

Author response: We agree with the reviewer that 6 M HCl would also target poorly crystalline FeS species (Heron et al, 1994)⁴⁷. We changed the text accordingly, see comment above.

Figure 4. I'm surprised there's such a remarkable decrease in organic C from the palsa to the bog and fen, especially given the increasingly water-logged conditions. Has this been examined? I would imagine from looking at these data that the total C stock decreases substantially.

Author response: We thank the reviewer for this comment. Yes, there is a decrease in total OC from palsa to fen as has been shown already (Siewert, 2018⁵⁶; Lupascu *et al.*, 2012³²). Although this only accounts for soil carbon and not carbon in the above ground biomass (e.g. cotton grass in the fen).

l. 246. Suggest "However, this estimate does not account for deeper layers."

Author response: We agree with the suggestion of the reviewer and changed the sentence accordingly, see also following comment (Discussion, page 16, line 280-281).

l. 247. This statement that deeper layers could contain even more Fe-bound C is a bit of a leap, especially since Fe-bound C seems to be concentrated in the transition layer in these soils. Perhaps it could introduce more Fe into the system, but that isn't really known.

Author response: We agree with the reviewer's comment, and thus deleted the statement and revised the text accordingly:

Page 16, line 280-284 (Discussion): "However, this estimate does not account for carbon sequestration by biomass in the fen or deeper layers. It is therefore crucial to further

determine the amount of iron in deeper layers, the amount of carbon bound to reactive iron minerals in numerous permafrost environments, and the lability/bioavailability of this carbon following its release.”

l. 267. Yes, your EXAFS data support the increase in the proportion of organic-bound Fe(II)!

l. 335. Check this statement – it is difficult to tell from the graphs because of the different axes, but it appears to me that Palsa B has a higher peak reactive Fe concentration than Palsa A.

Author response: The reviewer is correct, Palsa B has a higher amount of reactive Fe than Palsa A, but lower Fe-associated organic carbon than Palsa A. The text was revised accordingly in the main text, and the axis changed to make them clearer (maximum of 500 mg carbon per g soil) in the SI:

Page 21, line 393-396, page 20 (Material and Methods): “The long-term stored core Palsa B still showed higher abundance of reactive iron-associated organic carbon than Bog D and Fen F, but less than Palsa A which could be due to natural variability, long-term storage or because it was taken closer to the collapsing edge (Fig. S6 and Fig. S8).“

Page 18, line 218-219 (Supplementary Information):

1. 353. Does samples in this sentence refer to pore waters? Why were the pore waters centrifuged if they were already filtered?

Author response: We thank the reviewer for pointing out this mistake. These are the porewater samples which have been filtered by the rhizon sampler through 0.15 μm and thus have not been centrifuged. We corrected the text accordingly:

Page 22, line 412-417 (Material and Methods): "Rhizon porewater samplers (Rhizosphere research products, Netherlands) with a porous sampling area of 10 cm and 0.15 µm pore size were used to extract porewater from three different depths, resulting in one sample representing each organic horizon, transition zone and mineral horizon. [text deleted] The extracted porewater was analyzed for dissolved Fe (total and Fe(II)), organic carbon (DOC) and fatty acids. For total Fe and Fe(II), the supernatant was acidified in 1 M hydrochloric acid (HCl)..."

l. 376. The structure of this paragraph was a little confusing. I suggest revising as follows:

"Prior to use, all glassware was washed with 1 M HCl for 10 min, flushed three times with deionized water and once with MilliQ water. Afterwards glassware was sterilized at 180°C in the oven for 4.5 hours. 0.3 g dry soil was weighed into a 10 mL glass vial with 6.25 mL extractant and N₂ headspace. Following extraction, as detailed below, all samples were centrifuged at room temperature for 10 min at 5300 g."

Author response: We agree with the reviewer and revised the section as suggested:

Page 23, line 445-450 (Material and Methods): "Prior to use, all glassware was washed with 1 M HCl for 10 min, flushed three times with deionized water and once with MilliQ water. Afterwards glassware was sterilized at 180°C in the oven for 4.5 hours. 0.3 g dry soil was weighed into a 10 mL glass vial with 6.25 mL extractant and N₂ headspace. Following the extraction, as detailed below, all samples were centrifuged at room temperature for 10 min at 5300 g."

l. 391. I agree that the blank correction is important. Was the calibration curve also run in the matrix solution to check for interference?

Author response: We thank the reviewer for this important question. The samples were diluted 1:10 and 1:100 in 1M HCl under anoxic conditions to stabilize the Fe(II) even under oxic conditions. During the ferrozine assay, which is used to quantify Fe(II) and Fe(total) in a 96-well plate, the samples are further 1:100 diluted with the reaction solutions (hydroxylamine-HCl or HCl and ferrozine). Thus, the sample is finally diluted 1:1000 and 1:10,000. The standards for the ferrozine assay are also prepared in 1 M HCl, and our samples highly diluted in HCl, therefore we can comfortably say that our samples and standard are run in the same matrix. Given the high levels of dilution we are comfortable that no matrix effects of the extractant solution would remain.

Page 24, line 466-469 (Material and Methods): “At the same time, samples and standards are heavily diluted in 1 M HCl and hydroxylamine-HCl (1:1000 and 1:10,000) before spectrophotometric quantification. Given these high levels of dilution, no matrix effects of the extractant solution remains. “

l. 456. The EXAFS determinations were not clear to me. Were all reference spectra used in the linear combination fits? How did you decide which fit was the best? Were contributions rejected if they were below a certain value?

Author response: We thank the reviewer for pointing this out. Reference compounds were chosen based on prior knowledge of the sample mineralogy including, for example, criteria such as elemental composition, elements in the soil extracts of the different Fe phases, site characteristics (e.g. redox conditions, pH), and principal component analysis (PCA). Reference spectra were included in the fit only if they contributed an Fe fraction of 5 wt% or more. Detection limit for minor constituents is approximately 5 wt%. We now revised the suggested section accordingly:

Page 27, line 534-548 (Material and Methods): “A set of Fe reference compounds was used to perform linear combination fitting (LCF) of EXAFS spectra in SixPack from chi values of 2 to 12 with an x-weight of 3. Non-negative fits were performed. Reference compounds were chosen based on prior knowledge of the sample including, for example, criteria such as elemental composition (determined by element composition of the soil extracts), site characteristics (e.g. redox conditions, pH), and principal component analysis (PCA). PCA determines the number of distinct species in series of spectra. All contributions below 5 wt% were eliminated since we have previously determined that the limit of detection for mixed Fe species is around 5 wt% (Hansel *et al.*, 2003⁵⁷; Shimizu *et al.*, 2013⁸). We determined the best “least square fitting” based on fitting parameters such as the reduced chi² (X^2) and R-factor values. The best fits were reference samples such as natural nontronite and ferrosmectite (referred to as Fe clays) obtained from the Clay Mineral Society, Fe(II)-citrate and Fe(III)-citrate (referred to as Fe(II)-OM and Fe(III)-OM) which were prepared and analyzed as described in Daugherty *et al.* (2017)⁵⁸, mackinawite (referred to as FeS) which was prepared and analyzed as described in Troyer *et al.* (2014)⁵⁹ and 2-line ferrihydrite (referred to as poorly crystalline Fe), prepared and analyzed as described in Borch *et al.* (2007)⁶⁰ (Fig. 3).

Table S1. Unclear what OC and what Fe is represented in the OC:Fe ratio.

Author response: We thank the reviewer for clarifying this. The OC and Fe values used for the mass OC:Fe ratio are the concentrations of extractable Fe and OC (mg per g soil) obtained by the sodium dithionite citrate extraction, also presented in the table. We now changed Table S1 and the text referring to it.

Page 14-15, line 239-248 (Results): "...previously shown for intact permafrost soils³³.

The maximum mass ratio of organic carbon to iron of 0.22, based on the maximal sorption capacity of reactive iron oxides for natural organic matter (Wagai & Mayer, 2007¹⁰; Kaiser & Guggenberger, 2007⁶¹), as has been previously done to further characterize Fe-C associations in such system (Herndon *et al.*, 2017⁴⁵; Mu *et al.*, 2016⁴³), was exceeded for reactive Fe-associated organic carbon:reactive Fe, determined via the sodium dithionite citrate extraction, in the bulk palsa soil layers (organic/transition/mineral), in the bulk bog organic horizon and in the bulk bog transition zone (8.59±3.0 OC:Fe (wt:wt)) (Table S1). This suggests co-precipitation and/or chelation of organic compounds in the bulk sample which can generate structures with OC:Fe ratios above 0.22, as shown in other studies (Wagai & Mayer, 2007¹⁰). Existence of such structures is consistent with our high sodium pyrophosphate extractable iron values.

Page 2, line 2-11 (Supplementary Information):

	Reactive iron (control corrected)	Reactive iron of total extractable iron	Control iron	C bound to reactive iron (control corrected)	C bound to reactive iron of the total organic carbon	Control carbon	OC:Fe (wt:wt)	Total organic carbon	Total extractable Fe
	mg/g	%	mg/g	mg/g	%	mg/g		mg/g	mg/g
Palsa A									
Organic horizon	0.40 ± 0.11	100.00	0.00 ± 0.00	0.94 ± 0.58	0.22	1.37 ± 0.01	2.35	423 ± 0.00	0.20 ± 0.02
	0.29 ± 0.09	100.00	0.00 ± 0.00	2.16 ± 0.95	0.51	1.65 ± 0.08	7.45	422.91 ± 0.13	0.17 ± 0.00
Transition zone	2.55 ± 0.57	72.86	0.29 ± 0.08	30.99 ± 0.71	9.93	3.13 ± 0.02	12.15	312.11 ± 0.33	3.51 ± 0.08
	8.44 ± 0.21	93.86	0.75 ± 0.11	52.50 ± 0.13	14.80	10.36 ± 0.50	6.22	354.72 ± 0.04	8.99 ± 0.28
Mineral horizon	3.17 ± 0.19	36.58	0.25 ± 0.03	27.39 ± 1.61	20.13	2.88 ± 0.08	8.64	136.11 ± 0.21	8.65 ± 0.28
	1.35 ± 0.21	10.00	0.07 ± 0.06	13.58 ± 0.42	18.67	1.39 ± 0.10	10.06	72.71 ± 0.29	13.48 ± 0.22
Bog C									
Organic horizon	1.48 ± 0.18	40.60	0.73 ± 0.07	16.16 ± 3.91	4.85	3.16 ± 1.67	10.92	333.31 ± 0.05	3.63 ± 0.05
Transition zone	2.08 ± 0.05	11.14	0.41 ± 0.04	22.67 ± 8.60	39.42	1.18 ± 0.21	10.90	57.51 ± 0.38	18.65 ± 0.70
Mineral horizon	0.88 ± 0.06	7.52	0.28 ± 0.04	0.00 ± 0.00	0.00	1.04 ± 0.01	0.00	8.28 ± 0.25	11.69 ± 0.81
Fen E									
Organic horizon	2.03 ± 0.14	43.39	0.75 ± 0.00	0.00 ± 0.00	0.00	1.53 ± 0.00	0.00	234.70 ± 0.83	4.68 ± 0.01
Transition zone	2.64 ± 0.03	18.29	0.37 ± 0.00	0.00 ± 0.00	0.00	1.38 ± 0.18	0.00	16.24 ± 0.18	14.46 ± 0.22
Mineral horizon	1.75 ± 0.04	10.71	0.15 ± 0.00	0.00 ± 0.00	0.00	2.57 ± 0.76	0.00	3.52 ± 0.05	16.34 ± 0.44
	1.70 ± 0.04	8.95	0.19 ± 0.01	0.00 ± 0.00	0.00	1.13 ± 0.17	0.00	4.99 ± 0.10	19.01 ± 0.25

Table S1. Absolute and % values of iron and carbon in locations Palsa A, Bog C and Fen E, i.e. the cores reported in the main text. In most of the layers, the maximum mass ratio of organic carbon to iron (reactive Fe-associated organic carbon:reactive Fe) exceeds 0.22, the maximal sorption capacity of reactive iron oxides for natural organic matter (Wagai & Mayer, 2007¹⁰; Kaiser & Guggenberger, 2007⁶¹). Co-precipitation and/or chelation of organic compounds can generate structures with OC:Fe ratios (wt:wt)

above 0.22, as shown in other studies (Wagai & Mayer, 2007¹⁰). Errors indicate the range of duplicate analyses of each layer in each thaw stage.

Figure S8 (b and c). No organic horizon? Incorrect depths?

Author response: We again thank the reviewer for pointing this out and explained this now in the text. The depths are correct, the organic horizon was method-inherent lost during coring. A different type of corer was used for the triplicate cores (Pürckhauer corer), which is open on one side. The wet organic horizon was not solid enough to be retained by the corer and could not be collected. See changes as mentioned above.

Reviewer 3:

Patzner et al present an interesting study on iron- and organic carbon cycling in thawing permafrost soils. Many earlier studies only focused on the release of “more stable” organic carbon (OC) from iron oxide dissolution (e.g. doi: 10.4319/lo.1998.43.6.1287). This study now confirms that iron-bound DOC is partly bioavailable and shows how it is released.

The topic of the study is timely and important in the context of climate change and biogeochemistry. The many methods used to produce the collected data set are impressive. The results are interesting for readers from environmental microbiology, biogeochemistry, and to some extent for climate modelers. The study therefore merits publication.

The study is technically sound (with one exception) and can basically be published as is. I only have one major remark and very few marginal comments.

Author response: We thank the reviewer for this positive feedback. We carefully followed the suggestions provided by the reviewer and changed the manuscript accordingly.

Major remark:

One of the main hypothesis of this work is that thawing releases bioavailable DOC from iron oxides. However, as far as I can see, nothing was done to prevent the degradation of the bioavailable organic matter during the extraction experiments. The DOC concentrations released after dithionite extraction were corrected for DOC that is released by NaCl/NaHCO₃ of similar ionic strength. However, when shaking soil with water you create an ideal setting for microbial activity as you release large amounts of nutrients and OC while continuously shaking the mixture. Without defeating the undesired microbial consumption of OC you get up to 50 % less water-soluble OC per unit of soil (see figure 1a in doi: 10.1111/ejss.12256). This is a major problem of soil-water extracts, because the estimates of water-soluble organic carbon are underestimated without suppression of microbial activity. From the methods section it is not clear if microbial activity was suppressed during the experiments. If this was done and I missed it, please accept my apologies and nothing needs to be done. But if not, the estimates on water-soluble OC should be discussed. The numbers for NaCl/NaHCO₃ control extraction are usually low (Fig. S2) so the error here should not be too large. Extraction with 6 M HCL should certainly lead to complete disinfection. But for the dithionite extraction I am uncertain. It appears that dithionite has only minor ecotoxicological effects, so microbial activity is probably not suppressed by the addition of dithionite. Because the general conclusions of the study are not in danger, I think that a literature based discussion on this issue might be sufficient.

*Author response: We thank the reviewer for this very valuable comment! We agree with the reviewer that shaking water with soil has the potential to create a suitable setting for microbial activity, which could further affect the released organic carbon. We agree that under suitable settings (as for example in the suggested paper Riedel *et al.*, 2015⁶² which used an ionic strength of 0.02 M) that further controls (e.g. sodium azide addition) would be required to determine the extent of microbial activity during extraction. However, in*

our setup, we reach an ionic strength of 1.85 M in the dithionite-citrate bicarbonate extraction and the associated control extraction (sodium chloride bicarbonate), which is already above the optimum growth condition for slight halophiles and is more favorable for moderate halophiles (0.8 to 3.4 M salt content) (Ollivier *et al.*, 1994⁶³). Moderate to extreme halophiles were so far not found at the studied permafrost peatland which is a freshwater wetland (Woodcroft *et al.*, 2018³⁴). Rath *et al.* (2012)⁶⁴ previously showed that salinity exerts a strong inhibitory effect on a range of microbial processes in soil and that the acute toxic effects occurred immediately (within 2 h). We carefully followed the broadly applied extraction procedure, as has been used by others (Wagai & Mayer, 2007¹⁰ and Wagai *et al.*, 2013³⁹; Coward *et al.*, 2017¹⁷; Loeppert & Inskeep, 1996³⁸; Holmgren, 1967³⁷). We therefore believe that microbial activity during and after the dithionite citrate extraction can be excluded. As suggested by the reviewer, we now added a literature-based discussion in the manuscript, see accordingly:

Page 25-26, line 506-510 (Material and Methods): “Microbial activity during and after the dithionite-citrate extraction and trisodium citrate sodium bicarbonate extraction can be excluded due to high salt content (ionic strength of 1.85 M). Rath *et al.* (2012)⁶⁴ previously showed that salinity exerts a strong inhibitory effect on a range of microbial processes in soil and that acute toxic effects occurred immediately (within 2h).”

Minor remarks:

Line 25-27: This is probably only true for humid climates. In arid regions iron oxides typically only change upon aging (“Ostwald ripening”), which happens on times-scales of years or even centuries.

Author response: We thank the reviewer for his comment and changed the sentence accordingly:

Page 3, line 35-36 (Introduction): “The inventory of reactive iron minerals in humid climates is highly dynamic as they precipitate and dissolve in response to changing redox conditions.”

Lines 140-143: This is also repeated in the introduction and the methods, which may not be necessary. Please consider shortening the manuscript here by removing repetition.

Author response: We thank the reviewer for the comment. Because we are not limited in space and think that this short repetition helps the reader substantially, we left the text unchanged here.

Line 247-249: If this is true, than permafrost thawing is going to have a substantial impact from the release of iron-bound carbon alone. However, please bear in mind, that when a fen has evolved in the final stages of thawing it will act as a carbon sink again. This could be added here.

Author response: We thank the reviewer for this comment and changed the text accordingly.

Page 16, line 280-281 (Discussion): “However, this estimate does not account for carbon sequestration by biomass in the fen or deeper layers.”

Line 362-364 and 392-401: Information on precision is already included in the figures with the error bars. But some information on accuracy or trueness would be great such as “Accuracy was better than xx %”, if the data is available. If the NPOC mode for DOC analysis was used, how was inorganic carbon removed?

Author response: We thank the reviewer for this comment and agree that it would be beneficial to have information on accuracy or trueness of the measured Fe and organic carbon concentrations. We now changed the text accordingly:

Page 22, line 420-422 (Material and Methods): “For the DOC analysis, the calibration curves were $r^2 > 0.999$ and the standard deviations of the triplicate analysis were $<2\%$.”

Page 24, line 469-475 (Material and Methods): “For the ferrozine assay, the calibration curves were $r^2 > 0.999$, and the standard deviations of the triplicate analyses were $<1\%$. For the ICP-MS, the calibration curves were $r^2 > 0.999$, and the standard deviations of the triplicate analyses were $<5\%$. For the MP-AES, the calibration curves were $r^2 > 0.993$ and the standard deviations of the triplicate analysis were $<10\%$. The results of all Fe analysis (ferrozine assay, MP-AES/ICP-MS analysis) show all the same trends with depth and along the thaw gradient (S3). For additional extractant specific experimental parameters see below.”

Page 11, line 135-142 (Supplementary Information): “MP-AES was also used to determine aluminum concentrations in the extracts (A1) (Figure S4). For the ferrozine assay, the calibration curves were $r^2 > 0.999$, and the standard deviations of the triplicate analyses were $<1\%$. For the ICP-MS, the calibration curves were $r^2 > 0.999$, and the standard deviations of the triplicate analyses were $<5\%$. For the MP-AES, the calibration curves were $r^2 > 0.993$ and the standard deviations of the triplicate analysis were $<10\%$. We are aware of differences between the iron values. However, because the values only vary slightly, the ferrozine values had a higher accuracy ($<1\%$) and sodium dithionite citrate was not measured with ICP-MS and MP-AES due to citric formation after acidification, we decided to use the data from the ferrozine assay.”

Page 26, line 517-518 (Material and Methods): “For the TOC analysis, the calibration curves were $r^2 > 0.998$ and the standard deviations of the triplicate analysis were $<1\%$.”

The inorganic carbon was removed by acidifying with 16% HCl before the TOC analysis via combustion (see page 26, line 513-514) and by acidifying the samples with 2 M HCl before DOC analysis in the extracts to determine Fe-associated OC. We now revised the text accordingly:

Page 22, line 421-422 (Material and Methods): “Inorganic carbon was removed by acidifying the samples with 2 M HCl prior analysis.”

Line 393: Please spell out abbreviations (MP-AES/ICP-MS) and explain in a little more detail what was done. Which of the elements was measured by either of the methods? The information is given in the supplement text belonging to Fig. S3, but including it here in the methods section would be beneficial for the organization of the paper.

Author response: We agree with the reviewer and now spelled out the abbreviations, explained each analysis more in detail and added the measured elements by either of the methods.

Page 23-24, line 453-462 (Material and Methods): “Additionally, the samples were acidified in 1% (v/v) HNO₃ and analyzed in duplicates by MP-AES (microwave plasma atomic emission spectroscopy) or ICP-MS (inductively coupled plasma mass spectrometry). To get the total iron (Fe), phosphorous (P) and sulphur (S) concentrations, the extracts were analyzed using ICP-MS. To further obtain aluminum

(Al) concentrations and a cross-check of the total Fe concentrations, the MP-AES was used for analysis (S4 and S5). The illustrated iron values throughout the whole study represent the iron values obtained by the ferrozine assay...”

Page 13, line 159-163, Figure caption S5 (Supplementary Information): “Iron (Fe), phosphorous (P) and sulphur (S) concentrations were measured with ICP-MS. Iron (Fe) and aluminum (Al) concentrations were analyzed using MP-AES. The illustrated Fe values here are measured by MP-AES. The slightly different Fe concentrations by the different analytical approaches (ferrozine assay, MP-AES and ICP-MS) are shown in figure S3.”

Fig. 2-4: it might be easier for readers to understand the numbers when the units are similar (either as mg/l or as mM)...

Author response: We thank the reviewer for this comment and now added a second x-axis for each row, showing the porewater iron and organic carbon concentrations in mM at the top and in mg/L at the bottom. See changes as follows:

Page 9, line 137-138 (Results):”

Fig. 4b: I assume that upper, middle and lower pictures are always organic, transition and mineral horizon. Maybe this could be stated in the caption or indicated in the figure.

Author response: Exactly, we indicated the different horizons with green, grey and yellow background behind the nanoSIMS images, but made it now even more visible and mentioned it again in the figure caption. We thank the reviewer for pointing this out.

Fig. 4. Fe-C associations along the thaw gradient determined by bulk (b) and fine fraction analysis (a). a, Carbon bound by reactive iron minerals along the thaw gradient. The carbon which dislodged from the soil during the reductive dissolution of reactive iron oxides (orange) is shown in comparison to the total organic carbon determined via combustion (black grids, labeled as TOC). Dithionite-citrate extractable carbon is “control corrected” by subtracting the measured DOC content of a citrate solution and the measured DOC value from the NaCl control experiment. The NaCl control (same ionic strength and same pH as the sodium dithionite citrate extraction) shows negligible carbon release (Table S1, S2-S5). Errors indicate the range of duplicate analyses of each layer in each thaw stage. b, High spatial resolution analysis of iron-carbon associations by nanoSIMS along the thaw gradient (two end-members palsa (left) and fen (right)). The strong spatial association of C to Fe(III) minerals could only be observed in the palsa transition zone. The other fine fractions showed organic-free iron minerals. For the two end-members palsa and fen, four particles of the fine fractions of each layer were analyzed by nanoSIMS, all showing the same spatial distribution of Fe and C as shown by these six representatives (see also Fig. S9). The green background marks the organic horizon (b, upper images), grey the transition zone (b, middle images), and yellow the mineral horizon (b, lower images).”

Supplement text S3. Here the difference of the three methods used to measure iron is shown and discussed. Because reference material for such extraction done by the authors are not available it is difficult to obtain data on trueness and accuracy of each method. If you have no further

information then each of the methods is equally well suited from an analytical point of view. In this case one may simply calculate the mean of all values. However, this is just a suggestion and should not be considered mandatory during revision.

Author response: We thank the reviewer for carefully revising our manuscript and the valuable feedback. In the main text, we used the Fe values obtained by the ferrozine assay, because the data is available for all applied extractions (sodium dithionite citrate bicarbonate, sodium chloride bicarbonate, hydroxylamine HCl, sodium pyrophosphate, 6 M HCl). Due to citric acid formation after acidification, the sodium dithionite citrate extract was not measured with the more sensitive MP-AES and ICP-MS instrument, which means we cannot average the Fe values. The reason we added the Fe values for the extracts which are available (6 M HCl, hydroxylamine HCl and sodium pyrophosphate) to the SI was to show the consistency of trends through various analyses.

In all an interesting study that I like to see published eventually.

Thank you for considering me as a reviewer,

Thomas Riedel

Author response: Thank you for your feedback! Your comments improved the manuscript substantially.

References of Revision letter

- 1 Chen, C., Hall, S. J., Coward, E. & Thompson, A. Iron-mediated organic matter decomposition in humid soils can counteract protection. *Nat Commun* **11**, 2255 (2020).
- 2 Turetsky, M. R. *et al.* Short-term response of methane fluxes and methanogen activity to water table and soil warming manipulations in an Alaskan peatland. *J Geophys Res-Bioge* **113**, G000496 (2008).
- 3 Zona, D. *et al.* Methane fluxes during the initiation of a large-scale water table manipulation experiment in the Alaskan Arctic tundra. *Global Biogeochem Cy* **23**, GB2013 (2009).
- 4 Lipson, D. A., Jha, M., Raab, T. K. & Oechel, W. C. Reduction of iron (III) and humic substances plays a major role in anaerobic respiration in an Arctic peat soil. *J Geophys Res-Bioge* **115**, G00I06 (2010).
- 5 Herndon, E. M. *et al.* Pathways of anaerobic organic matter decomposition in tundra soils from Barrow, Alaska. *J Geophys Res-Bioge* **120**, 2345-2359 (2015).
- 6 Olefeldt, D., Turetsky, M. R., Crill, P. M. & McGuire, A. D. Environmental and physical controls on northern terrestrial methane emissions across permafrost zones. *Global Change Biol* **19**, 589-603 (2013).
- 7 Perryman, C. R. *et al.* Thaw Transitions and Redox Conditions Drive Methane Oxidation in a Permafrost Peatland. *J Geophys Res-Bioge* **125**, G005526 (2020).
- 8 Shimizu, M. *et al.* Dissimilatory Reduction and Transformation of Ferrihydrite-Humic Acid Coprecipitates. *Environ Sci Technol* **47**, 13375-13384 (2013).
- 9 Estop-Aragones, C. *et al.* Limited release of previously-frozen C and increased new peat formation after thaw in permafrost peatlands. *Soil Biol Biochem* **118**, 115-129 (2018).
- 10 Wagai, R. & Mayer, L. M. Sorptive stabilization of organic matter in soils by hydrous iron oxides. *Geochim Cosmochim Acta* **71**, 25-35 (2007).
- 11 Kogel-Knabner, I. *et al.* Organo-mineral associations in temperate soils: Integrating biology, mineralogy, and organic matter chemistry. *J Plant Nutr Soil Sc* **171**, 61-82 (2008).
- 12 Lalonde, K., Mucci, A., Ouellet, A. & Gelinas, Y. Preservation of organic matter in sediments promoted by iron. *Nature* **483**, 198-200 (2012).
- 13 Kleber, M. *et al.* Mineral-Organic Associations: Formation, Properties, and Relevance in Soil Environments. *Adv Agron* **130**, 1-140 (2015).
- 14 Asano, M. & Wagai, R. Evidence of aggregate hierarchy at micro- to submicron scales in an allophanic Andisol. *Geoderma* **216**, 62-74 (2014).
- 15 Totsche, K. U. *et al.* Microaggregates in soils. *J Plant Nutr Soil Sc* **181**, 104-136 (2018).
- 16 Riedel, T., Zak, D., Biester, H. & Dittmar, T. Iron traps terrestrially derived dissolved organic matter at redox interfaces. *P Natl Acad Sci USA* **110**, 10101-10105 (2013).
- 17 Coward, E. K., Thompson, A. T. & Plante, A. F. Iron-mediated mineralogical control of organic matter accumulation in tropical soils. *Geoderma* **306**, 206-216 (2017).
- 18 Henneberry, Y. K., Kraus, T. E. C., Nico, P. S. & Horwath, W. R. Structural stability of coprecipitated natural organic matter and ferric iron under reducing conditions. *Org Geochem* **48**, 81-89 (2012).
- 19 Eusterhues, K. *et al.* Reduction of ferrihydrite with adsorbed and coprecipitated organic matter: microbial reduction by *Geobacter bremensis* vs. abiotic reduction by Na-dithionite. *Biogeosciences* **11**, 4953-4966 (2014).
- 20 Coward, E. K., Thompson, A. & Plante, A. F. Contrasting Fe speciation in two humid forest soils: Insight into organomineral associations in redox-active environments. *Geochim Cosmochim Acta* **238**, 68-84 (2018).

- 21 Roden, E. E. *et al.* Extracellular electron transfer through microbial reduction of solid-phase humic substances. *Nat Geosci* **3**, 417-421 (2010).
- 22 Jones, A. M., Collins, R. N., Rose, J. & Waite, T. D. The effect of silica and natural organic matter on the Fe(II)-catalysed transformation and reactivity of Fe(III) minerals. *Geochim Cosmochim Acta* **73**, 4409-4422 (2009).
- 23 Amstatter, K., Borch, T. & Kappler, A. Influence of humic acid imposed changes of ferrihydrite aggregation on microbial Fe(III) reduction. *Geochim Cosmochim Acta* **85**, 326-341 (2012).
- 24 Masue-Slowey, Y., Loeppert, R. H. & Fendorf, S. Alteration of ferrihydrite reductive dissolution and transformation by adsorbed As and structural Al: Implications for As retention. *Geochim Cosmochim Acta* **75**, 870-886 (2011).
- 25 Hall, S. J., Berhe, A. A. & Thompson, A. Order from disorder: do soil organic matter composition and turnover co-vary with iron phase crystallinity? *Biogeochemistry* **140**, 93-110 (2018).
- 26 ThomasArrigo, L. K., Byrne, J. M., Kappler, A. & Kretzschmar, R. Impact of Organic Matter on Iron(II)-Catalyzed Mineral Transformations in Ferrihydrite-Organic Matter Coprecipitates. *Environ Sci Technol* **52**, 12316-12326 (2018).
- 27 Jilling, A. *et al.* Minerals in the rhizosphere: overlooked mediators of soil nitrogen availability to plants and microbes. *Biogeochemistry* **139**, 103-122 (2018).
- 28 Williams, E. K., Fogel, M. L., Berhe, A. A. & Plante, A. F. Distinct bioenergetic signatures in particulate versus mineral-associated soil organic matter. *Geoderma* **330**, 107-116 (2018).
- 29 Van Bodegom, P. M., Scholten, J. C. M. & Stams, A. J. M. Direct inhibition of methanogenesis by ferric iron. *FEMS Microbiol Ecol* **49**, 261-268 (2004).
- 30 Wagner, R., Zona, D., Oechel, W. & Lipson, D. Microbial community structure and soil pH correspond to methane production in Arctic Alaska soils. *Method Enzymol* **19**, 3398-3410 (2017).
- 31 Malhotra, A. & Roulet, N. T. Environmental correlates of peatland carbon fluxes in a thawing landscape: do transitional thaw stages matter? *Biogeosciences* **12**, 3119-3130, (2015).
- 32 Lupascu, M., Wadham, J. L., Hornibrook, E. R. C. & Pancost, R. D. Temperature Sensitivity of Methane Production in the Permafrost Active Layer at Stordalen, Sweden: a Comparison with Non-permafrost Northern Wetlands. *Arct Antarct Alp Res* **44**, 469-482 (2012).
- 33 Hodgkins, S. B. *et al.* Changes in peat chemistry associated with permafrost thaw increase greenhouse gas production. *P Natl Acad Sci USA* **111**, 5819-5824 (2014).
- 34 Woodcroft, B. J. *et al.* Genome-centric view of carbon processing in thawing permafrost. *Nature* **560**, 49-54 (2018).
- 35 IPCC, 2007: Summary for policy makers. In Solomon, S., Qin, D., Manning, M., Chen, Z., Marquis, M., Averyt, K. B., Tignor, M., and Miller, H. L. (eds.), *Climate Change 2007: The Physical Science Basis. Contribution of Working Group I to the Fourth Assessment Report of the Intergovernmental Panel on Climate Change*. Cambridge and New York: Cambridge University Press, 1-18.
- 36 Olefeldt, D. & Roulet, N. T. Effects of permafrost and hydrology on the composition and transport of dissolved organic carbon in a subarctic peatland complex. *J Geophys Res-Biogeophys* **117**, G01005 (2012).
- 37 Holmgren, G. G. A Rapid Citrate-Dithionite Extractable Iron Procedure. *Soil Sci Soc Am Pro* **31**, 210-211 (1967).
- 38 Loeppert, R. H. & Inskeep, W. Iron. In: Sparks, D.L., Page, A.L., Helmke, P.A., Loeppert, R.H., Soltanpour, P.N., Tabatabai, M.A., Johnston, C.T., Summer, M.E. (1996).

- 39 Wagai, R., Mayer, L. M., Kitayama, K. & Shirato, Y. Association of organic matter with iron and aluminum across a range of soils determined via selective dissolution techniques coupled with dissolved nitrogen analysis. *Biogeochemistry* **112**, 95-109 (2013).
- 40 Ryan, J. N. & Gschwend, P. M. Extraction of iron oxides from sediments using reductive dissolution by titanium(III). *Clays Clay Min* **39**, 509-518 (1991).
- 41 Mehra, O. P. J., M. L. Iron oxide removal from soils and clays by a dithionite-citrate system buffered with sodium bicarbonate. *Clays Clay Min* **7**, 317-327 (1958).
- 42 Kleber, M., Mikutta, R., Torn, M. S. & Jahn, R. Poorly crystalline mineral phases protect organic matter in acid subsoil horizons. *Eur J Soil Sci* **56**, 717-725 (2005).
- 43 Mu, C. C. *et al.* Soil organic carbon stabilization by iron in permafrost regions of the Qinghai-Tibet Plateau. *Geophys Res Lett* **43**, 10286-10294 (2016).
- 44 Pokrovsky, O. S., Manasypov, R. M., Loiko, S. V. & Shirokova, L. S. Organic and organo-mineral colloids in discontinuous permafrost zone. *Geochim Cosmochim Acta* **188**, 1-20 (2016).
- 45 Herndon, E. *et al.* Influence of iron redox cycling on organo-mineral associations in Arctic tundra soil. *Geochim Cosmochim Acta* **207**, 210-231 (2017).
- 46 Poulton, S. W. & Canfield, D. E. Development of a sequential extraction procedure for iron: implications for iron partitioning in continentally derived particulates. *Chem Geol* **214**, 209-221 (2005).
- 47 Heron, G., Crouzet, C., Bourg, A. C. M. & Christensen, T. H. Speciation of Fe(I) and Fe(II) in Contaminated Aquifer Sediments Using Chemical-Extraction Techniques. *Environ Sci Technol* **28**, 1698-1705 (1994).
- 48 Aller, R. C., Mackin, J. E. & Cox, R. T. Diagenesis of Fe and S in Amazon Inner Shelf Muds - Apparent Dominance of Fe Reduction and Implications for the Genesis of Ironstones. *Cont Shelf Res* **6**, 263-289 (1986).
- 49 Widdel, F., Kohring, G. W. & Mayer, F. Studies on Dissimilatory Sulfate-Reducing Bacteria That Decompose Fatty-Acids .3. Characterization of the Filamentous Gliding Desulfonema-Limicola Gen-Nov Sp-Nov, and Desulfonema-Magnum Sp-Nov. *Arch Microbiol* **134**, 286-294 (1983).
- 50 Widdel, F. & Pfennig, N. Studies on Dissimilatory Sulfate-Reducing Bacteria That Decompose Fatty-Acids .1. Isolation of New Sulfate-Reducing Bacteria Enriched with Acetate from Saline Environments - Description of Desulfobacter-Postgatei Gen-Nov, Sp-Nov. *Arch Microbiol* **129**, 395-400 (1981).
- 51 Widdel, F. Anaerobic degradation of fatty acids and benzoic acid by newly isolated species sulphate-reducing bacteria. Dissertation, Universität Göttingen, FRG (1980).
- 52 Straub, K. L., Kappler, A. & Schink, B. Enrichment and isolation of ferric-iron- and humic-acid-reducing bacteria. *Method Enzymol* **397**, 58-77 (2005).
- 53 Laufer, K. *et al.* Coexistence of Microaerophilic, Nitrate Reducing, and Phototrophic Fe(II) Oxidizers and Fe(III) Reducers in Coastal Marine Sediment. *Appl Environ Microbiol* **82**, 3694-3694 (2016).
- 54 Cornish, E. A. & Fisher, R. A. Moments and cumulants in the specification of distributions. *Revue de l'Institut International de Statistique/Rev. Int. Stat. Inst.* **5** 4, 307-320 (1938).
- 55 Salama, I. A. K., G. G.; Tolley, H.D. . On the estimation of the most probable number in a serial dilution technique. . *Commun. Stat. - Theory Methods* **A7**, 1267-1281 (1978).
- 56 Siewert, M. B. High-resolution digital mapping of soil organic carbon in permafrost terrain using machine learning: a case study in a sub-Arctic peatland environment. *Biogeosciences* **15**, 1663-1682 (2018).

- 57 Hansel, C. M. *et al.* Secondary mineralization pathways induced by dissimilatory iron reduction of ferrihydrite under advective flow. *Geochim Cosmochim Acta* **67**, 2977-2992 (2003).
- 58 Daugherty, E. E., Gilbert, B., Nico, P. S. & Borch, T. Complexation and Redox Buffering of Iron(II) by Dissolved Organic Matter. *Environ Sci Technol* **51**, 11096-11104 (2017).
- 59 Troyer, L. D., Tang, Y. & Borch, T. Simultaneous reduction of arsenic(V) and uranium(VI) by mackinawite: role of uranyl arsenate precipitate formation. *Environ Sci Technol* **48**, 14326-14334 (2014).
- 60 Borch, T., Masue, Y., Kukkadapu, R. K. & Fendorf, S. Phosphate imposed limitations on biological reduction and alteration of ferrihydrite. *Environ Sci Technol* **41**, 166-172 (2007).
- 61 Kaiser, K. & Guggenberger, G. Sorptive stabilization of organic matter by microporous goethite: sorption into small pores vs. surface complexation. *Eur J Soil Sci* **58**, 45-59 (2007).
- 62 Riedel, T., Hennessy, P., Iden, S. C. & Koschinsky, A. Leaching of soil-derived major and trace elements in an arable topsoil after the addition of biochar. *Eur J Soil Sci* **66**, 823-834 (2015).
- 63 Ollivier, B., Caumette, P., Garcia, J. L. & Mah, R. A. Anaerobic-Bacteria from Hypersaline Environments. *Microbiol Rev* **58**, 27-38 (1994).
- 64 Rath, K. M., Maheshwari, A., Bengtson, P., Rousk, J. Comparative toxicities of salts on microbial processes in soil *Applied and Environmental Microbiology* **82**, 2012-2020 (2012).

REVIEWERS' COMMENTS

Reviewer #1 (Remarks to the Author):

The authors have done a thorough job addressing my concerns and I find the manuscript to be of suitable significance and broad interest for publication in Nature Communications.

I would add one minor point, that the newly published study of Fisher et al. (2020) provides new evidence that the citrate-bicarbonate-dithionite extraction method results in only partial dissolution of the most susceptible FeR phase (ferrihydrite) during OC-Fe extractions, and therefore incomplete removal of bound OC.

This is relevant because the authors now point out that OC-Fe associations are more resistant to microbial or chemical reductions (line 33 'Several studies already identified poorly crystalline Fe organic matter associations in the field or produced them in the lab, and demonstrated that they are resistant to microbial or chemical reduction (Henneberry et al., 2012; Eusterhues et al., 2014, Coward et al., 2018)').

Together these aspects of OC-Fe associations indicate that OC sequestration by reactive Fe minerals is likely to be more significant than currently estimated.

Given the introductory rationale for the authors' work at lines 23 - 35, the newly published findings of Fisher et al., would help strengthen this narrative.

Reviewer #2 (Remarks to the Author):

Thank you to the authors for addressing my concerns and those of the other reviewers. The inclusion of data from the 2019 cores, which were analyzed using suitably comparable methods to the 2018 cores presented in the main text, provides enough reasonable replication to support the conclusions. I am now supportive of publishing this manuscript in Nature Communications. I have a few more comments for the authors to consider at their discretion:

- 1) It's fine for the authors to represent soil depth in the figures, but they should state this with the associated caveats. As written, the text only says that they do not report depth ("the data is presented by different layers rather than depth"). The clarification regarding the lack of organic horizon in certain cores is appreciated.
- 2) It's great that the authors calculated Fe stocks, although they are not discussed in the main text?
- 3) I think the authors misunderstood my question regarding the significance of the transition layer – I was not asking broadly about the significance of the study. I was asking why the Fe and Fe-bound C are concentrated in the transition layer between the organic and mineral horizons.

Congrats!

Reviewer #3 (Remarks to the Author):

Dear Colleagues,

the manuscript has undergone substantial revisions. The authors have done an excellent job in addressing

the reviewers concerns.
In my opinion the article is now ready to be published.

Best regards,
Thomas Riedel

Response to Reviewers' Comments

We thank the three reviewers and the editor, Dr. Kyle Frischkorn, for their positive feedback and their final evaluations of our manuscript titled “*Iron mineral dissolution releases iron and associated organic carbon during permafrost thaw*” (manuscript ID: NCOMMS-20-20682-T). We are grateful to the reviewers for their favorable review of our work and the constructive comments and suggestions that significantly improved the quality of this manuscript. We appreciate the opportunity to publish in Nature Communication and have carefully taken every comment into consideration and revised and edited our manuscript accordingly.

In this revision letter, the reviewers' comments are shown in italic font, our responses are in normal font and green. We highlighted the changes we made in the manuscript in blue in this letter. Please note that all line numbers refer to the track changes version of the revised manuscript (i.e., with all changes included). Below is the point by point response to the reviewers' comments.

Reviewer 1:

The authors have done a thorough job addressing my concerns and I find the manuscript to be of suitable significance and broad interest for publication in Nature Communications.

I would add one minor point, that the newly published study of Fisher et al. (2020) provides new evidence that the citrate-bicarbonate-dithionite extraction method results in only partial dissolution of the most susceptible FeR phase (ferrihydrite) during OC-Fe extractions, and therefore incomplete removal of bound OC.

This is relevant because the authors now point out that OC-Fe associations are more resistant to microbial or chemical reductions (line 33 'Several studies already identified poorly crystalline Fe organic matter associations in the field or produced them in the lab, and demonstrated that they are resistant to microbial or chemical reduction (Henneberry et al., 2012; Eusterhues et al., 2014, Coward et al., 2018)').

Together these aspects of OC-Fe associations indicate that OC sequestration by reactive Fe minerals is likely to be more significant than currently estimated.

Given the introductory rationale for the authors' work at lines 23 - 35, the newly published findings of Fisher et al., would help strengthen this narrative.

Author response: We would like to thank the reviewer for this positive feedback and valuable last comment. We agree with the reviewers comment that the new findings of this study indicate that OC sequestration by reactive Fe minerals is likely to be more significant than currently estimated. Therefore, we now included the newly published study of Fisher *et al.* in the manuscript, see the following changes:

Page 16, line 291-294 (Discussion): “Fisher et al. (2020) even showed evidence for only partial dissolution of the Fe phases and associated organic carbon by the dithionite-citrate approach, indicating that carbon sequestration by Fe minerals is likely to be more significant than currently estimated.”

Reviewer 2:

Thank you to the authors for addressing my concerns and those of the other reviewers. The inclusion of data from the 2019 cores, which were analyzed using suitably comparable methods to the 2018 cores presented in the main text, provides enough reasonable replication to support the conclusions. I am now supportive of publishing this manuscript in Nature Communications. I have a few more comments for the authors to consider at their discretion:

Author response: We would like to thank the reviewer for supporting our manuscript for publishing in Nature Communications. We have done our best to further address the reviewer's findings. Details are listed below.

1) It's fine for the authors to represent soil depth in the figures, but they should state this with the associated caveats. As written, the text only says that they do not report depth ("the data is presented by different layers rather than depth"). The clarification regarding the lack of organic horizon in certain cores is appreciated.

Author response: We thank the reviewer for this comment. We now changed the text accordingly:

Page 21, line 422-426 (Material and Methods): "Hammering caused slight compaction of the cores, mainly in the organic horizon. Therefore, the recorded depths are not completely comparable to the initial soil profiles. Thus, the data is presented by the different soil layers (organic horizon, transition zone and mineral horizon) and depth intervals are still reported to illustrate that the dataset represents the active layer depth along the thaw gradient."

2) It's great that the authors calculated Fe stocks, although they are not discussed in the main text?

Author response: We thank the reviewer for this comment. We referred to the calculated Fe stocks in the main text (page 10, line 79-81, Results) and report them in the SI, but due to slight compaction in the organic horizon, we are not confident in discussing the stocks and prefer to discuss Fe and C contents. We still did some final

edits and mentioned the stock calculation in the Conclusion, see following changes in the main text:

Page 16, line 305-308: “Stock calculations (Supplementary Figure 1) show a higher Fe-associated OC fraction in mineral horizon than in the palsa transition zone which could indicate that OC sequestration by reactive Fe minerals is likely to be more significant in deeper layers than currently estimated.”

3) *I think the authors misunderstood my question regarding the significance of the transition layer – I was not asking broadly about the significance of the study. I was asking why the Fe and Fe-bound C are concentrated in the transition layer between the organic and mineral horizons.*

Author response: We thank the reviewer for clarifying. We now highlighted in text why we think the Fe and Fe-bound C are concentrated in the transition layer between the organic and mineral horizon. See following changes in the text:

Page 11, line 91-95 (Results): “We suggest that this is driven by Fe(III) reduction in deeper layers leading to mobilization of $\text{Fe}^{2+}_{\text{aq}}$. This can be subsequently re-oxidized close to the interface between the organic horizon and mineral horizon by O_2 which diffuses from the surface. Oxidation will result in precipitation as Fe(III) oxyhydroxide minerals at this transition, as has been previously suggested for boreal peat soils³⁴.”

Congrats!

Reviewer 3:

Dear Colleagues,

the manuscript has undergone substantial revisions.

The authors have done an excellent job in addressing the reviewers concerns.

In my opinion the article is now ready to be published.

*Best regards,
Thomas Riedel*

Author response: We thank the reviewer for this positive feedback.